# LLM-WRAPPER: BLACK-BOX SEMANTIC-AWARE ADAPTATION OF VISION-LANGUAGE MODELS FOR REFERRING EXPRESSION COMPREHENSION

**Amaia Cardiel**[1,2], **Éloi Zablocki**[1], **Elias Ramzi**[1], **Oriane Siméoni**[1], **Matthieu Cord**[1,3]

[1] Valeo.ai     [2] APTIKAL, LIG, Université Grenoble Alpes     [3] Sorbonne Université

`{amaia.cardiel, eloi.zablocki, elias.ramzi, matthieu.cord}@valeo.com`

## ABSTRACT

Vision Language Models (VLMs) have demonstrated remarkable capabilities in various open-vocabulary tasks, yet their zero-shot performance lags behind task-specific fine-tuned models, particularly in complex tasks like Referring Expression Comprehension (REC). Fine-tuning usually requires 'white-box' access to the model's architecture and weights, which is not always feasible due to proprietary or privacy concerns. In this work, we propose `LLM-wrapper`, a method for 'black-box' adaptation of VLMs for the REC task using Large Language Models (LLMs). `LLM-wrapper` capitalizes on the reasoning abilities of LLMs, improved with a light fine-tuning, to select the most relevant bounding box matching the referring expression, from candidates generated by a zero-shot black-box VLM. Our approach offers several advantages: it enables the adaptation of closed-source models without needing access to their internal workings, it is versatile as it works with any VLM, it transfers to new VLMs and datasets, and it allows for the adaptation of an ensemble of VLMs. We evaluate `LLM-wrapper` on multiple datasets using different VLMs and LLMs, demonstrating significant performance improvements and highlighting the versatility of our method. While `LLM-wrapper` is not meant to directly compete with standard white-box fine-tuning, it offers a practical and effective alternative for black-box VLM adaptation. The code and the checkpoints are available at `https://github.com/valeoai/LLM_wrapper`.

## 1 INTRODUCTION

Vision Language Models (VLMs), a class of foundation models (Bommasani et al., 2021), trained on large-scale and diverse tasks and datasets, have shown remarkable abilities to solve various open-vocabulary tasks, as image captioning (Xiao et al., 2024; Li et al., 2023), visual question answering (Alayrac et al., 2022; Liu et al., 2023; Li et al., 2023), text-image retrieval (Radford et al., 2021; Zhai et al., 2023; Li et al., 2023), object detection (Liu et al., 2024c; Xiao et al., 2024; Cheng et al., 2024), or semantic segmentation (Ding et al., 2023; Xiao et al., 2024). Recent VLMs show promising zero-shot generalization abilities to new tasks and data domains (Wei et al., 2022; Alayrac et al., 2022). However, there is still a significant performance gap between zero-shot VLMs and those that have been specifically trained or adapted for a particular task and data domain. This work focuses on the challenging open-vocabulary detection task of Referring Expression Comprehension (REC) (Mao et al., 2016), which involves localizing an object in an image based on a complex textual query, requiring both spatial and semantic reasoning. While zero-shot VLMs can typically detect most objects mentioned in the query with reasonably accurate bounding boxes and labels, they struggle to identify *only* the described object. Moreover, VLMs often have difficulty understanding complex descriptions that involve relations between objects, attributes, or negations (Xie et al., 2023; Yao et al., 2024).

To improve performance, VLMs are typically fine-tuned on the specific REC task and corresponding datasets. This fine-tuning is usually done in a *'white-box'* manner, with full access to the model's architecture and weights for back-propagation. However, this process requires expertise to design fine-tuning objectives and optimize hyper-parameters, specific to each VLM and downstream task. Moreover, white-box fine-tuning is not always feasible. Some models are closed-source, either because they are proprietary and released behind APIs (Ren et al., 2024a), or because they are trained on private data, making their weights and gradients inaccessible. While companies may provide APIs for adapting proprietary models, e.g., (OpenAI, 2024), these solutions are limited to predefined scopes and require sharing data with external private companies, raising legal and privacy concerns.

Table 1: **Comparison between white-box fine-tuning and `LLM-wrapper` on the REC task.**

|  | **Classic white-box fine-tuning** | **`LLM-wrapper`** |
|---|---|---|
| Model access | Needs complete access (loss, architecture, weights, gradients) | Only needs a forward call of a frozen black-box VLM |
| Fine-tuning knowledge | Specific for each VLM | Agnostic to the choice of VLM and LLM, and parameter-efficient |
| Specificity of adaptation | None | Leverages LLM semantic reasoning |
| Fine-tuning generalization | Limited: no ensembling, no transfer to new VLMs or datasets | Flexible: supports ensembling and transfers across VLMs and datasets |
| Expected performances | Best results | Good results |

To address these challenges, we explore *'black-box'* adaptation of VLMs, where only forward calls to the model are possible. We propose `LLM-wrapper`, a new method for the black-box adaptation of VLMs for the REC task, by using an LLM to reason on the VLM's outputs. This approach builds on the recent development of Large Language Models (LLMs) (Touvron et al., 2023; Jiang et al., 2024; Mesnard et al., 2024), which have shown interesting reasoning capabilities. The underlying idea is that a zero-shot black-box VLM can generate high-quality labeled bounding boxes, and that `LLM-wrapper` can then leverage the semantic and spatial reasoning abilities of the LLM (Lian et al., 2024a;b) to 'reason' on such outputs, further enhanced with a light fine-tuning. As illustrated in Figure 1, our method involves translating VLM's outputs into a natural language prompt and feeding it to the LLM. The LLM is then tasked with identifying the box that best matches the referring expression from the given candidates.

`LLM-wrapper` offers several advantages, summarized in Table 1. First, since the adaptation is done in a black-box manner, it removes the need for back-propagation through the VLM and allows for the adaptation of closed-source models, without requiring access to the model's architecture, weights, or gradients. This makes black-box fine-tuning versatile and easy to use, as it can be applied to any model without requiring specific model knowledge or assumptions about the model's architecture. Additionally, as the adaptation is delegated to the LLM via text, `LLM-wrapper` retains the pre-trained knowledge of the original VLM. We find that `LLM-wrapper` transfers well to other VLMs, generalizes to model updates, and can adapt effectively to new datasets without additional fine-tuning. Finally, `LLM-wrapper` enables the adaptation of an ensemble of VLMs, leveraging the flexibility of text-based adaptation to handle varying numbers of bounding boxes, thereby combining strengths from multiple models.

We experiment with `LLM-wrapper` on three classic REC datasets — RefCOCO, RefCOCO+ (Kazemzadeh et al., 2014), RefCOCOg (Mao et al., 2016) — and on Talk2Car (Deruyttere et al., 2019), using two notably different VLMs — Grounding-DINO (Liu et al., 2024c) and Florence-2 (Xiao et al., 2024). Additionally, we evaluate `LLM-wrapper` on the recent and challenging HC-RefLoCo (Wei et al., 2024) benchmark. We also experiment with two LLMs for the adaptation: Mixtral 8x7B (Jiang et al., 2024) and Llama 3 8B (Dubey et al., 2024). While `LLM-wrapper` is not meant to outperform standard white-box fine-tuning, we show that `LLM-wrapper` significantly enhances the VLM's performances, across all combinations of VLMs, LLMs and datasets, thus demonstrating the versatility of our method. Notably, on RefCOCOg and Talk2Car, our most challenging benchmarks with respect to semantic understanding, `LLM-wrapper` improves the results of zero-shot VLMs with gains ranging from +8.7 up to +18.3 P@1.

## 2 RELATED WORK

**Referring Expression Comprehension (REC) and Vision Language Models (VLMs).** Referring Expression Comprehension (REC) (Kazemzadeh et al., 2014; Mao et al., 2016; Qiao et al., 2021) is the task of identifying objects in an image based on referring expressions. Typically, REC involves selecting the best region from a set of region proposals extracted from the image, guided by a referring expression. This task is challenging because the referring expression can range from a short phrase (Kazemzadeh et al., 2014) to a long sentence that may require multi-step reasoning (Mao et al., 2016). REC is distinct from similar tasks such as visual grounding (Rohrbach et al., 2016), where multiple object regions described by multiple noun phrases must be localized in an image. It also differs from object detection (Girshick, 2015), which uses predefined categories instead of natural language expressions. The recent surge of interest in VLMs, starting from CLIP (Radford et al., 2021) to the recent Florence-2 (Xiao et al., 2024) has significantly improved performance on tasks requiring both vision and language. For REC, several approaches have

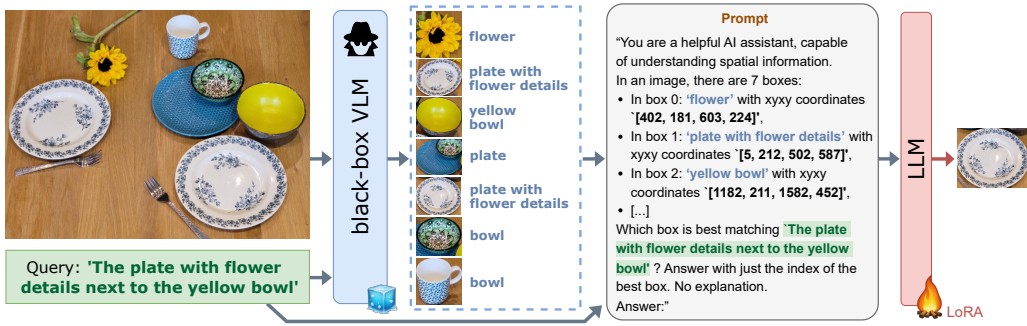

Figure 1: **Illustration of `LLM-wrapper`**. Our method adapts a black-box VLM for the REC task. The VLM output is translated into natural language to prompt an LLM. The latter is tasked with identifying the box that best matches the query among the given candidates. The LLM must learn to identify the query's subject and to disambiguate the correct object from distractors (e.g., several plates with 'flower details').

used VLMs. For instance, Grounding-DINO (Liu et al., 2024c) relies on multiple stages of modality fusion to align visual and textual features, while Florence-2 (Xiao et al., 2024) is a sequence-to-sequence model trained on a huge collection of data. However, despite being trained on extensive amounts of image-text data (e.g., 126 million images with 500 million to 3.6 billion annotations for Florence-2), these models' zero-shot performances on REC are sub-optimal compared to their performances when including REC data in their training set or when fine-tuned (see Section 4.2). Our approach, `LLM-wrapper`, addresses this clear limitation in a black-box manner, without the need to re-train the VLM with task-specific data.

**VLM adaptation.** While retraining the entire VLM on a new task or data domain is computationally expensive (Liu et al., 2023; Wang et al., 2023; Chen et al., 2023; Xiao et al., 2024), fine-tuning offers a more efficient alternative. However, even fine-tuning can be costly. To address this, parameter-efficient fine-tuning, e.g., LoRa (Hu et al., 2022), DoRa (Liu et al., 2024a), or VeRa (Kopiczko et al., 2024), and soft prompt (Li & Liang, 2021) learning have been proposed. These methods avoid updating the model's pre-trained weights but require access to the model's architecture and gradients.

A few recent methods propose strategies to adapt VLMs when back-propagation through the VLM is not feasible (Ouali et al., 2023; Yu et al., 2023; Liu et al., 2024b; Oh et al., 2023; Wan et al., 2024). Both Ouali et al. (2023) and Yu et al. (2023) focus on CLIP-based methods and adapt the VLM by learning either a feature projection or soft prompts. They however require access to the inner representations, which is generally not possible with APIs. Liu et al. (2024b) optimize the input prompt template, but also focuses on CLIP-based models. Instead, we target all VLMs that perform open-vocabulary detection, e.g., Grounding-DINO and Florence-2. Oh et al. (2023) adapt a VLM in a black-box manner by modifying the image input. However, this method is designed to address visual domain gaps and is not directly applicable to adapt a VLM to a new *task*. Moreover, it requires multiple API calls for each training image. Finally, Wan et al. (2024) introduce an adaptation method for frozen VLMs, applicable to REC. For each candidate bounding box, they perform a VLM inference to compute the probability of the input text based on a modified image where the box has been blacked out. The candidate that achieves the highest probability contrast with respect to the unmodified input image is chosen. Though training-free, this approach requires the access to the model's output probability distribution and performs an additional VLM inference for each candidate bounding box. In this work, we propose a strategy to adapt VLMs, and more specifically open-vocabulary detectors, to a new task, in a complete black-box fashion. Our method does not require access to the model's intermediate representations, output distribution, or gradients, runs a single inference/API call for each (text-image) input pair, and enables adaptation to a new task (REC in our case) that the VLM was not originally trained for.

## 3 `LLM-WRAPPER`: BLACK-BOX SEMANTIC-AWARE ADAPTATION OF VLMS.

We present `LLM-wrapper`, a novel LLM-based approach to adapt VLMs for the REC task. We detail the general idea in Section 3.1, the prompt construction in Section 3.2 and the LLM fine-tuning in Section 3.3.

### 3.1 GENERAL IDEA

Our method *wraps* the open-vocabulary detections of a frozen black-box VLM with an LLM that reasons over these outputs. An overview is presented in Figure 1. Given a complex textual query, `LLM-wrapper` leverages the fact that detection-oriented VLMs can typically localize well most nouns of the query, even if they struggle with the reasoning step required to precisely select the object of interest among several distractors. Therefore, `LLM-wrapper` delegates the reasoning task to an LLM, which has interesting abilities to handle difficult text queries, including attributes, negation, and relational or spatial descriptions of objects. As the gradients of the black-box VLM are not accessible, we propose to adapt its *outputs* with the LLM. It gives us access to the LLM gradients and we can thus specialize the LLM for the task, with a simple and light fine-tuning, performed on the outputs of the VLM to learn to select the right box among them. Overall, `LLM-wrapper` only requires black-box access to the VLM, whereas standard fine-tuning strategies need white-box access to perform back-propagation. This makes our approach more flexible and applicable in scenarios where the internal workings of the VLM are not accessible.

### 3.2 PROMPT CONSTRUCTION

The key idea of our method is to convert the VLM's outputs into natural language. To achieve this, we list all predicted outputs in the LLM prompt, including their box coordinates, labels, and, when applicable, prediction scores (displayed below in light gray). This allows the 'blind' LLM, which only reads text, to understand the scene and reason about the image. The prompt then reminds the query and asks the LLM to select the best matching box. For instance, given the query (in green in Figure 1), and the associated outputs (e.g., 'flower', 'plate with flower details', 'yellow bowl', etc.), we ask the LLM for the best matching box index, as follows:

> You are a helpful AI assistant, capable of understanding spatial information.
> In an image, there are 7 boxes:
> * In box 0: 'flower' with xyxy coordinates '[402, 181, 603, 224]' with score 0.92,
> * In box 1: 'plate with flower details' with xyxy coordinates '[5, 212, 502, 587]' with score 0.88,
> * In box 2: 'yellow bowl' with xyxy coordinates '[1182, 211, 1582, 452]' with score 0.86,
> * [...]
> Which box is best matching 'The plate with flower details next to the yellow bowl' ?
> Answer with just the index of the best box. No explanation.
> Answer:

### 3.3 FINE-TUNING THE LLM

While a zero-shot LLM can already reason on the new task to some extent, we find that fine-tuning the LLM significantly improves performances on the task. Therefore, we fine-tune the LLM for prompt completion with a cross-entropy loss, using the prompt described above. Specifically, the expected answer for the LLM is the index of the best box proposal, corresponding to the closest match to the known ground truth box. To build the training dataset, we use the REC training data, which consists of `(image, query)` pairs and ground truth boxes. The detection outputs (boxes, labels, scores), used to create the training prompts, are inferred using the VLM being adapted. We only keep the samples where at least one of the VLM box proposals has an Intersection over Union (IoU) with the ground truth box higher than 0.5, ensuring no noisy samples. To make `LLM-wrapper` robust to any shortcut learning based on the box order, we randomly permute the order of the box proposals in the prompt during training.

We leverage the extensive literature on LLM fine-tuning to specialize the LLM of `LLM-wrapper`. Specifically, we use LoRA (Hu et al., 2022), which introduces additive updates to the model's activations, parameterized by low-rank modules. Doing so reduces the number of new parameters to learn while preserving the LLM's general knowledge. We also use flash attention (Dao et al., 2022) and 4-bit quantization (Dettmers et al., 2024), making `LLM-wrapper` trainable on a single 40GB-A100 GPU in less than 7 hours. This approach makes the training efficient in terms of compute and very simple to implement in practice. Overall, we find that `LLM-wrapper` is not very sensitive to the choice of the few hyper-parameters introduced (see Section 4.4).

# 4 EXPERIMENTS

In this section we present the experimental validation of `LLM-wrapper`. We first state our protocol in Section 4.1. In Section 4.2, we show how `LLM-wrapper` improves VLMs. We then conduct further analysis of `LLM-wrapper`'s benefits in Section 4.3. We finally conduct ablation studies showcasing `LLM-wrapper`'s robustness in Section 4.4.

## 4.1 EXPERIMENTAL SETUP AND TECHNICAL DETAILS

### 4.1.1 REFERRING EXPRESSION COMPREHENSION (REC) TASK

**Datasets.** We evaluate `LLM-wrapper` on the REC task. In REC, given an input pair `(image, query)`, a model is expected to predict a single bounding box around the object described in the query, as illustrated in Figure 1. We use three standard datasets for REC: RefCOCO, RefCOCO+ (Kazemzadeh et al., 2014), and RefCOCOg (Mao et al., 2016) which is more challenging as it contains longer descriptions (8.3 words in average). We also use the driving, non human-centric, Talk2Car (Deruyttere et al., 2019) REC dataset. Additionally, we evaluate `LLM-wrapper` on HC-RefLoCo (Wei et al., 2024) for zero-shot dataset transfer experiments. All of these datasets contain multiple distractor objects for the given text query. Dataset statistics are given in Table 2.

**Metric.** We measure performance with the standard precision@1 (P@1) metric, described in (Qiao et al., 2021). A true positive is defined when the predicted box has an Intersection-Over-Union (IoU) greater than 0.5 with the ground truth box. The metric is averaged over the evaluation set.

Table 2: **Dataset statistics.**

| Dataset | Split used | Size train | val | test | # words / query |
|---|---|---|---|---|---|
| RefCOCO | unc | 120,624 | 10,834 | 10,752 | 3.5 |
| RefCOCO+ | unc | 120,191 | 10,758 | 10,615 | 3.5 |
| RefCOCOg | umd | 80,512 | 4,896 | 9,602 | 8.3 |
| Talk2Car | — | 8,348 | 1,163 | 2,447 | 11.0 |
| HC-RefLoCo | — | — | 13,360 | 31,378 | 84.6 |

### 4.1.2 THE VLMS

We evaluate the impact of `LLM-wrapper` on two different VLMs. In all cases, we use the official model checkpoints and unless specified otherwise, we use the model versions that are not fine-tuned for the REC task, meaning that the models have not been exposed to any of the RefCOCO/+/g and Talk2Car datasets. Using the model setup described below, we feed from 2 to 45 boxes to `LLM-wrapper` per prompt.

**Grounding-DINO (GD)** (Liu et al., 2024c) aligns visual queries with text through stages of modality fusion and contrastive learning. Initially designed for open-vocabulary grounding, the model produces 900 bounding boxes, achieving a high recall but an under-performing precision on the REC task. For REC, we use `GD` by selecting the bounding box with the highest score relative to any token in the query. However, this method sometimes selects boxes based on query parts unrelated to the main object, such as other nouns in the sentence. To address this, we introduce Grounding-DINO-REC (`GDrec`), a more targeted approach for REC. `GDrec` identifies the query's subject, defined as the first noun group detected by SpaCy's dependency parser (Honnibal et al., 2020), and selects the bounding box that scores best against it. `GDrec` outperforms `GD` on RefCOCO/+/g datasets, and particularly on RefCOCOg, where the noun identification is more challenging, with e.g., +9.1 P@1 (test) for zero-shot models as shown in Table 3. To ensure a rather short prompt, we limit the number of box proposals by setting the box confidence score threshold to 0.15 for `GDrec` and to 0.2 for `GD`. `GD` / `GDrec`, not fine-tuned, use a SwinT (`T`) backbone while the fine-tuned versions are based on SwinB (`B`), as they are the only publicly available models. We also run experiments on a subset of RefCOCOg using Grounding-DINO 1.5 (Ren et al., 2024a) (`GD-1.5`), a recent detector behind API (online at: `Grounding-DINO-1.5-API`), which provides 300 free API calls. `GD-1.5` extends `GD` with a larger backbone, ViT-L (Fang et al., 2024), and training dataset.

**Florence-2 (`Flo2`)** (Xiao et al., 2024) is a sequence-to-sequence multi-task model. We use the Florence-2 Large version from Hugging Face. It is composed of a DaViT vision encoder (Ding et al., 2022) and a multi-modal encoder-decoder. `Flo2` can be prompted to deal with different grounding tasks. To evaluate `Flo2` on the REC task without adaptation, we use the box predicted for the 'open vocabulary detection' task. For `LLM-wrapper`'s adaptation of `Flo2`, we keep and concatenate the boxes predicted for both the 'open vocabulary detection' and the 'phrase grounding' task modes.

Table 3: **Main results of `LLM-wrapper` on the REC task**, in P@1↑, on RefCOCO/+/g and Talk2Car datasets. '`(T)`' and '`(B)`' stand for the '`SwinT`' and '`SwinB`' backbones respectively.

| | Model | | | RefCOCOg | | RefCOCO | | RefCOCO+ | | Talk2Car | |
|---|---|---|---|---|---|---|---|---|---|---|---|
| Adaptation | access | VLM | LLM | val-umd | test-umd | val-unc | test-unc | val-unc | test-unc | val | test |
| ∅ (zero-shot) | | GD(T) | | 60.09 | 59.32 | 50.69 | 50.94 | 51.65 | 51.79 | 55.37 | 58.44 |
| *Fine-tuning* | *White-box* | *GD(B)* | | *78.51* | *77.99* | *83.86* | *84.12* | *73.46* | *73.46* | *N/A* | *N/A* |
| LLM-wrapper | Black-box | GD(T) | Mixtral | 77.57 ↑17.5 | 77.05 ↑17.7 | 74.61↑23.9 | 73.46↑22.5 | 60.32↑8.7 | 60.08↑8.3 | 64.75 ↑9.4 | 67.14 ↑8.7 |
| LLM-wrapper | Black-box | GD(T) | Llama3 | 78.12 ↑18.0 | 77.36 ↑18.0 | 74.78↑24.1 | 73.98↑23.0 | 64.18↑12.5 | 63.82↑12.0 | 65.95 ↑10.6 | 68.61 ↑10.2 |
| ∅ (zero-shot) | | GDrec(T) | | 67.61 | 68.37 | 51.82 | 52.12 | 53.28 | 53.16 | 47.64 | 51.04 |
| *Fine-tuning* | *White-box* | *GDrec(B)* | | *80.19* | *79.85* | *83.84* | *84.21* | *73.68* | *73.58* | *N/A* | *N/A* |
| LLM-wrapper | Black-box | GDrec(T) | Mixtral | 78.47 ↑10.9 | 77.92 ↑9.6 | 72.61↑20.8 | 71.48↑19.4 | 63.79↑10.5 | 63.69↑10.5 | 63.89 ↑16.3 | 66.78 ↑15.7 |
| LLM-wrapper | Black-box | GDrec(T) | Llama3 | 78.25 ↑10.6 | 78.01 ↑9.6 | 73.97↑22.2 | 73.07↑21.0 | 64.13↑10.9 | 64.08↑10.9 | 63.97 ↑16.3 | 66.45 ↑15.4 |
| ∅ (zero-shot) | | Flo2 | | 67.91 | 66.16 | 55.94 | 57.21 | 53.31 | 54.26 | 46.78 | 47.53 |
| *Fine-tuning* | *White-box* | *Flo2* | | *90.32* | *91.02* | *93.07* | *93.42* | *88.19* | *88.49* | *N/A* | *N/A* |
| LLM-wrapper | Black-box | Flo2 | Mixtral | 78.96 ↑11.1 | 77.69 ↑11.5 | 68.85↑12.9 | 68.21↑11.0 | 57.58↑4.3 | 58.26↑4.0 | 61.65 ↑14.9 | 65.14 ↑17.6 |
| LLM-wrapper | Black-box | Flo2 | Llama3 | 78.76 ↑10.9 | 78.03 ↑11.9 | 71.74↑15.8 | 71.91↑14.7 | 62.63↑9.3 | 62.73↑8.5 | 61.74 ↑15.0 | 65.84 ↑18.3 |

### 4.1.3 THE LLMs AND THEIR FINE-TUNING

Our main experiments are conducted using two different LLMs: Mixtral 8x7B Instruct (Jiang et al., 2024) (v0.1) and Llama 3 8B Instruct (AI@Meta, 2024) with Hugging Face's implementation. We use Hugging Face's supervised fine-tuning pipeline (SFT) (HuggingFace, 2024), that allows to implement the training choices discussed in Section 3. Specifically, for a same LoRA's rank $r$, we train 352M parameters for Llama 3 8B (which is 4.20% of the model original size) and 114M parameters for Mixtral 8x7B (which is 0.24% of its original size). To further study the impact of the LLM scale, we also experiment with two additional families of models, Gemma 2 (Gemma, 2024) and GPT-Neo (Gao et al., 2021). The latter is a class of LLMs, based on a replication of the GPT-3 architecture, trained on the large curated Pile dataset (Gao et al., 2021), and which was not 'instructed' (Ouyang et al., 2022). We train `LLM-wrapper` with Adam (Kingma, 2014), with a batch-size of four, until convergence. We discuss in Section 4.4 the sample efficiency of `LLM-wrapper`. Unless stated otherwise, we use a learning rate of $10^{-5}$ and a rank of $r = 128$ for LoRA. These hyper-parameters work consistently across four datasets, two LLMs and two VLMs, demonstrating the robustness of `LLM-wrapper`. We study the performance robustness of `LLM-wrapper` for different hyper-parameters in Section 4.4. We provide ablations quantifying the impact of trainable parameters count in Appendix C and statistics on invalid LLM outputs in Appendix D.

### 4.2 `LLM-WRAPPER` CAN ADAPT VLMs TO THE REC TASK

In Table 3, we report the performances of VLMs on the REC task in three settings: zero-shot off-the-shelf VLM, after classic white-box fine-tuning, and after black-box adaptation with `LLM-wrapper`. The results are obtained by adapting the VLM using exclusively the training data specific to each benchmark.

Our first observation confirms that while VLMs demonstrate remarkable zero-shot performance on new tasks and datasets, their performance is still significantly lower than models specifically adapted to the given task (white-box fine-tuning). As shown in Table 3, for GD, GDrec and `Flo2`, there is a notable gap in performance, ranging from -11.5 to -37.1 P@1, between the zero-shot and fine-tuned versions.

Interestingly, we observe that for all combinations of VLMs and LLMs, `LLM-wrapper` can adapt VLMs to the new REC task, despite the black-box setting. For instance, on RefCOCOg, `LLM-wrapper` brings improvements over the zero-shot models ranging from +9.6 to +18.0 P@1. Notably, while our proposed variant GDrec outperforms GD by large margins in a zero-shot setting (e.g., +9.1 P@1-test) due to better subject identification, both models perform similarly when adapted with `LLM-wrapper`. This shows that `LLM-wrapper` particularly improves VLMs that lack abilities useful for the task, such as subject identification for GD. On RefCOCO, `LLM-wrapper` improves again by significant margins zero-shot VLMs, e.g., +23.0 P@1 (test) for GD with Llama 3 8B. Given that textual queries in RefCOCO are very short (3.5 words on average), subject identification is relatively easy. Therefore, these gains indicate that `LLM-wrapper` greatly helps to disambiguate the object of interest from the distractors. On RefCOCO+, the improvement brought by `LLM-wrapper` is lower but still significant (between +4.0 and +12.5 P@1). This is expected as RefCOCO+ is designed to exclude location words, and most referring expressions are thus purely appearance-based descriptions (Qiao et al., 2021) where `LLM-wrapper` brings less value for adaptation. On the driving Talk2Car dataset, `LLM-wrapper`'s results remain consistent with those observed on the classic human-centric RefCOCO/+/g benchmarks, with score boosts provided for all VLM/LLM combinations, ranging from +8.7 up to +18.3 P@1.

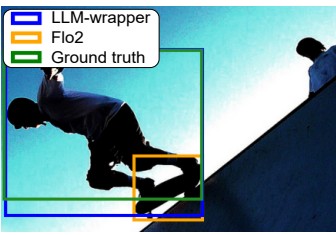 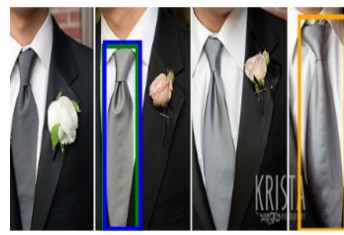 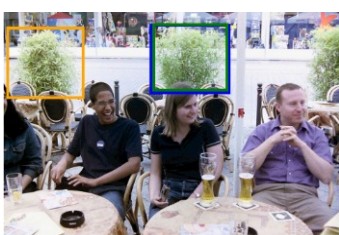

(a) *"Person on the skateboard"*    (b) *"The tie at the second from the left"*    (c) *"Green plant behind a table visible behind a lady's head"*

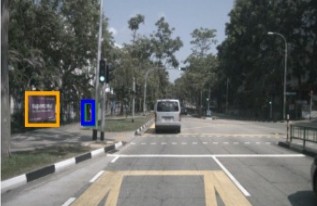 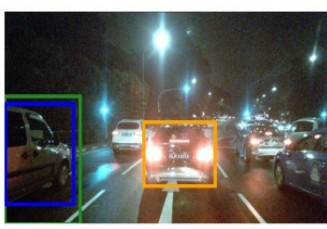 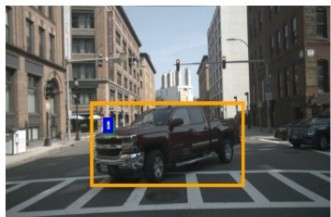

(d) *"She said something about a big sign on a fence. Maybe that is her! Pull over here by this person and we will find out"*    (e) *"Try to get in front of the car that passed us on the left. He is driving like a madman"*    (f) *"My friend said she would be standing on the corner waiting for me, I think that might be her, will you stop there?"*

Figure 2: **Qualitative results of `Flo2` on RefCOCOg (first row) and Talk2Car (second row), before and after adaptation with `LLM-wrapper`**, provided with queries as captions. Adapting `Flo2` (in **orange**) with `LLM-wrapper` (using Llama 3 8B, in **blue**) leads to improved reasoning and box selection.

We present some qualitative results in Figure 2, where we show predictions from `Flo2` before and after adaptation with `LLM-wrapper`, using Llama 3 8B. We observe that adaptation with `LLM-wrapper` enables better subject identification (Fig. 2a, Fig. 2d), spatial understanding (Fig. 2b, Fig. 2e), relational reasoning (Fig. 2a, Fig. 2c) and avoids selecting a more visible object when the ground truth is comparatively small (Fig. 2d, Fig. 2f). We display additional qualitative results, including failure cases, in Appendix E.

`LLM-wrapper` is not designed to outperform classic white-box fine-tuning but demonstrates competitive performance in some settings. For example, with `GD` and `GDrec` on RefCOCOg, `LLM-wrapper` achieves comparable results despite using a smaller vision backbone. Additionally, `LLM-wrapper` can complement white-box fine-tuning by adapting VLMs already optimized for the REC task. As shown in Appendix A, applying `LLM-wrapper` on top of fine-tuned models does not degrade performances and, in some cases, provides a slight boost. This highlights `LLM-wrapper`'s compatibility with state-of-the-art methods and its ability to enhance existing adaptations. Finally, in Appendix B, we extend our evaluation to the related task of Referring Expression Segmentation, leading to consistent score boosts.

## 4.3 BENEFITS OF WRAPPING VLMS' OUTPUTS WITH `LLM-WRAPPER`

**Ensembling VLMs.** One advantage of `LLM-wrapper` is its free-text input format, which allows adapting to various inputs. Indeed, we show that `LLM-wrapper` can learn to *ensemble* outputs from different VLMs, as shown in Table 4. Specifically, we concatenate the predictions of the two VLMs in the prompt described in Section 3. The results show that when ensembling `GDrec` and `Flo2`, scores are boosted by

Table 4: **Results of VLMs ensembling using `LLM-wrapper`** with Llama 3 8B, in P@1↑ on RefCOCOg using 'umd' splits. (Comparable findings for Mixtral).

| Adaptation | VLM | P@1 - val ↑ | P@1 - test ↑ |
|---|---|---|---|
| ∅ (zero-shot VLM) | GDrec | 67.61 | 68.37 |
| ∅ (zero-shot VLM) | Flo2 | 67.91 | 66.16 |
| LLM-wrapper | GDrec | 78.25 | 78.01 |
| LLM-wrapper | Flo2 | 78.76 | 78.03 |
| LLM-wrapper | Flo2 + GDrec | 81.25 | 80.13 |

Query: *"A bottle of wine between the vegetables"*

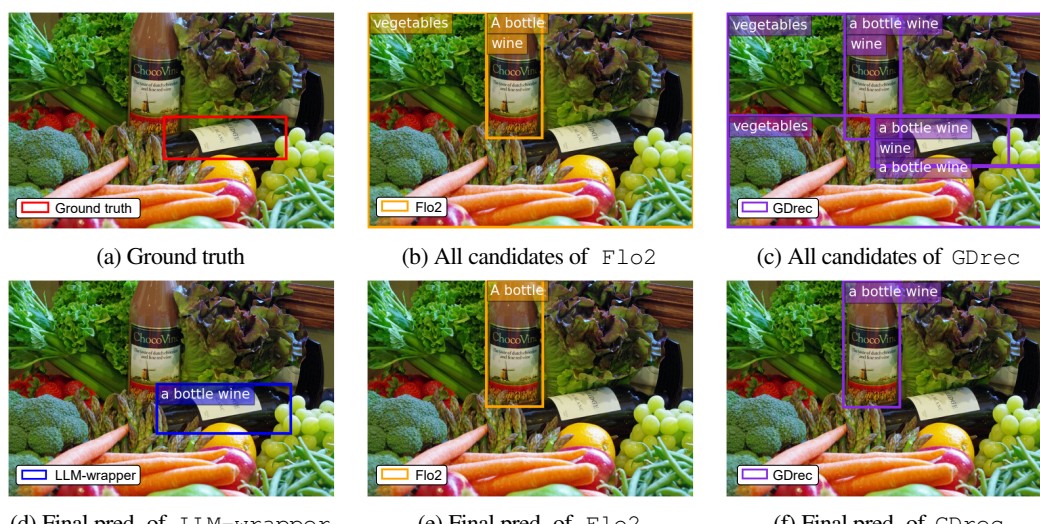

(a) Ground truth     (b) All candidates of `Flo2`     (c) All candidates of `GDrec`

(d) Final pred. of `LLM-wrapper`     (e) Final pred. of `Flo2`     (f) Final pred. of `GDrec`

Figure 3: **Visualizations of the candidates and predictions** for the query "A bottle of wine between the vegetables". We visualize the ground truth (a) and the set of box candidates generated by `Flo2` (b) and `GDrec` (c). In the second row, we visualize the final predictions of `Flo2` (e) and `GDrec` (f) and in (d) the prediction of `LLM-wrapper` applied on the ensemble of both VLMs' outputs, using Llama 3 8B. We observe that `LLM-wrapper` discards the distractor bottle and selects the correct object.

+2.5 P@1 (val-umd) and +2.1 P@1 (test-umd) when compared to those obtained with the best-performing VLM adapted with `LLM-wrapper`, namely `Flo2`. This demonstrates that `LLM-wrapper` is capable of reasoning on multiple sources and leveraging the strengths of different models.

Indeed, we observe that `Flo2` has a high precision, while `GD` and `GDrec` have lower precision but high recall. We show in Figure 3 a qualitative example where `LLM-wrapper` leverages the complementarity of the two models. The figure displays predictions from `Flo2` (Fig. 3e), `GDrec` (Fig. 3f), and `LLM-wrapper` ensembling predictions from `Flo2` and `GDrec` (Fig. 3d). We observe that while `Flo2` fails to detect the target object as a possible candidate (Fig. 3b), the additional proposals from `GDrec` (Fig. 3c) enable `LLM-wrapper` to find the correct object (Fig. 3a) that each independent model missed.

**Transferring a trained `LLM-wrapper` to a new VLM.** Another advantage of using text as the input format is that `LLM-wrapper` does not rely on model-specific activation values, making it transferable from one VLM to another. For instance, `LLM-wrapper` can be fine-tuned on `Flo2` and transferred to `GD` or `GDrec`. This is illustrated in Table 5, where, for instance, when fine-tuned on `GDrec`'s or `Flo2`'s outputs, transferring it at inference time to the other model's outputs gives an increase from +5.1 to +6.9 P@1 over zero-shot VLMs. This shows that during fine-tuning, `LLM-wrapper` learns spatial and semantic notions that generalize to other models. This capacity of `LLM-wrapper` is particularly useful for private models, such as `GD-1.5` (Ren et al., 2024a), where creating the training set can be expensive – for instance, getting predictions for RefCOCOg train would cost ≈ $1,600 ($20 per 1,000 API

Table 5: **Results of `LLM-wrapper` when using different VLMs' outputs during fine-tuning and inference,** in P@1↑ on RefCOCOg using 'umd' splits. Results obtained with Llama 3 8B. Comparable findings for Mixtral. [†]Scores obtained on a subset of 300 samples from RefCOCOg val-umd.

| Adaptation | VLM (fine-tuning) | VLM (inference) | P@1 - val ↑ (subset 300) | P@1 - val ↑ (full) | P@1 - test ↑ (full) |
|---|---|---|---|---|---|
| ∅ (zero-shot VLM) | ∅ | GDrec | 66.00[†] | 67.61 | 68.37 |
| LLM-wrapper | Flo2 | GDrec | 74.00[†] ↑**8.0** | 73.90 ↑**6.3** | 73.45 ↑**5.1** |
| ∅ (zero-shot VLM) | ∅ | Flo2 | 71.67[†] | 67.91 | 66.16 |
| LLM-wrapper | GDrec | Flo2 | 75.33[†] ↑**3.7** | 73.86 ↑**6.0** | 73.03 ↑**6.9** |
| ∅ (zero-shot VLM) | ∅ | GD-1.5 | 47.67[†] | — | — |
| LLM-wrapper | GDrec | GD-1.5 | 76.67[†] ↑**29.0** | — | — |

Table 6: **`LLM-wrapper` performance on zero-shot dataset transfer**. Results obtained with Llama 3 8B. 'FT' stands for 'fine-tuning'. With [†], `Flo2` is fine-tuned using 'a collection of public supervised data on a wide range of downstream tasks', including but not limited to RefCOCO/+/g (Xiao et al., 2024).

| Adaptation | VLM | Fine-tuning Data | Inference Data | P@1 - val ↑ | P@1 - test ↑ |
|---|---|---|---|---|---|
| ∅ (zero-shot VLM) | `Flo2` | — | HC-RefLoCo | 48.04 | 47.39 |
| *White-box FT* | *`Flo2`* | *RefCOCO/+/g[†]* | *HC-RefLoCo* | *56.75* ↑**8.7** | *55.62* ↑**8.2** |
| `LLM-wrapper` | `Flo2` | RefCOCOg | HC-RefLoCo | 66.93 ↑**18.9** | 66.45 ↑**19.1** |
| ∅ (zero-shot VLM) | `Flo2` | — | RefCOCO | 55.94 | 57.21 |
| `LLM-wrapper` | `Flo2` | RefCOCO | RefCOCO | 71.74 ↑**15.8** | 71.91 ↑**14.7** |
| `LLM-wrapper` | `Flo2` | RefCOCOg | RefCOCO | 69.00 ↑**13.1** | 68.88 ↑**11.7** |
| ∅ (zero-shot VLM) | `Flo2` | — | RefCOCO+ | 53.31 | 54.26 |
| `LLM-wrapper` | `Flo2` | RefCOCO+ | RefCOCO+ | 62.63 ↑**9.3** | 62.73 ↑**8.5** |
| `LLM-wrapper` | `Flo2` | RefCOCOg | RefCOCO+ | 61.00 ↑**7.7** | 61.07 ↑**6.8** |

calls). To illustrate this use case, we use a RefCOCOg val subset corresponding to 300 free API calls to `GD-1.5`. When `LLM-wrapper` is fine-tuned on `GDrec`'s outputs and applied on `GD-1.5`'s outputs for inference, results are boosted by a significant +29.0 P@1. To ensure that the val-subset reflects the full val set's difficulty, we evaluate `GDrec` and `Flo2` on both sets. The results are consistent, confirming that the val-subset is a good proxy for the full set and that the performance gains from `LLM-wrapper` on GD-1.5 are significant This showcases how `LLM-wrapper` can be used on private models or on continuously updated versions of models without the need to re-train it with each model update.

**Transferring a trained `LLM-wrapper` to new datasets.** `LLM-wrapper` demonstrates strong generalization across datasets, reducing the need for training data and resources on new target domains. To demonstrate this property, we adapt `Flo2` with `LLM-wrapper` on RefCOCOg and evaluate it on RefCOCO and RefCOCO+. Results in Table 6 show substantial gains over zero-shot `Flo2`, with improvements up to +13.1 P@1 on RefCOCO and +7.7 P@1 on RefCOCO+. Although slightly below results from direct fine-tuning on target datasets, these improvements demonstrate `LLM-wrapper`'s ability to transfer knowledge effectively. To test generalization to more complex scenarios, we evaluate `Flo2` adapted with `LLM-wrapper` using RefCOCOg on HC-RefLoCo, a benchmark with no training split and longer, more complex, referring expressions, with an average length of 84 words. `LLM-wrapper` achieves +18.9 and +19.1 P@1 improvements on the validation and test sets, respectively, compared to zero-shot `Flo2`. Interestingly, when white-box fine-tuned `Flo2` is transferred to HC-RefLoCo, its performance boost over zero-shot `Flo2` is less than half of that achieved by `LLM-wrapper`. These results highlight `LLM-wrapper`'s ability to handle in a zero-shot setting complex referring expressions despite being fine-tuned on ten times shorter expressions. More details on the impact of input complexity on performance are given in Appendix C.2. Qualitative examples on HC-RefLoCo are shown in Appendix E.1.

## 4.4 ABLATION STUDIES

Unless specified otherwise, we run ablations on RefCOCOg val-umd, using `GD`'s outputs and Llama 3 8B.

**Impact of LLM scale.** In Figure 4a, we report the P@1 REC performance ($y$-axis) for various LLM sizes ($x$-axis). We observe that `LLM-wrapper` is effective across all model families, and that the gain in performance positively correlates with the LLM size. Indeed, though the sub-billion model, GPT-Neo 139M, slightly boosts the P@1 performance of `GD` zero-shot (+2.2 P@1), this increase is only incremental when compared to Llama 3 8B's gains (+18.0 P@1). Further analysis shows a Pearson correlation of 0.88 between the LLM's original performance on reasoning benchmarks, measured by the HellaSwag score (Zellers et al., 2019), and REC performance (details in Appendix C.1). Overall, these findings demonstrate that larger, more capable LLMs are substantially more effective, while smaller models offer only limited improvements.

**Robustness to hyper-parameters.** In Figure 4b, we show that `LLM-wrapper` is not overly sensitive to the value of the LoRA rank, which controls the number of fine-tuned parameters. In the explored range, $r \in \{12, 64, 128, 192\}$, corresponding to fine-tuning $\{0.41\%, 2.15\%, 4.20\%, 6.18\%\}$ of total parameters, the performance varies by only $\pm 0.4$ P@1. We provide a more in-depth analysis of the impact of the number of fine-tuned parameters on performance in Appendix C.1. Similarly, we observe in Figure 4c that `LLM-wrapper` is loosely sensitive to the learning rate choice for values in $\{10^{-6}, 5 \times 10^{-6}, 10^{-5}\}$. With learning rate values higher that $10^{-5}$, we observe unstable training behaviors.

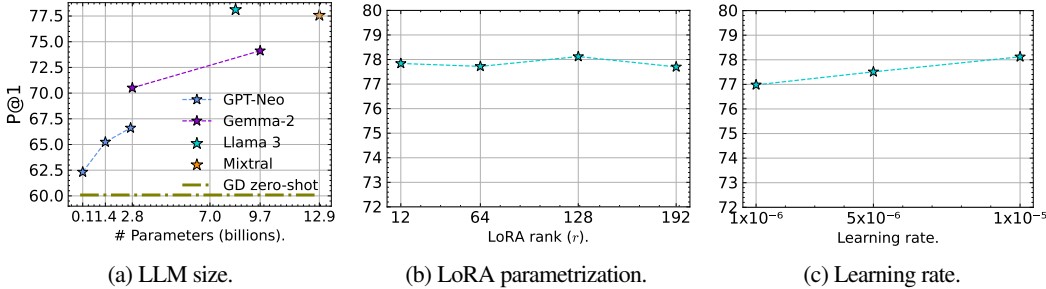

| (a) LLM size. | (b) LoRA parametrization. | (c) Learning rate. |

Figure 4: **Study of `LLM-wrapper`'s sensitivity to hyper-parameters.** Impact on `LLM-wrapper`'s performances (with `GD` on RefCOCOg) of the (a) LLM scale; (b) LoRA rank $r$; and (c) learning rate.

**Ablation on the number of training samples.** Figure 5 displays the P@1 REC performance on RefCOCOg val with respect to the number of training samples, when fine-tuning Llama 3 8B on RefCOCOg train. It shows a sharp increase in P@1 for all VLMs (most impressive for `GD`) during the first 30k samples, which takes ∼2h of training in our setting. Thus, even with a restricted amount of samples, `LLM-wrapper` can boost performances. In the extreme case of no training samples for the LLM, i.e., a zero-shot LLM, we observe on Table 7 that in most settings the VLM and the VLM + zero-shot LLM have very similar results. This follows the observation that, while LLMs have an extensive general knowledge, they may lack off-the-shelf reasoning (Kazemi et al., 2023; Fu et al., 2024).

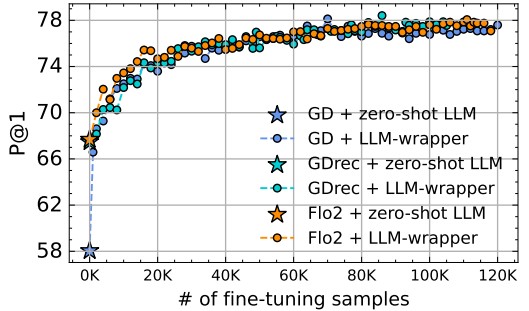

|  | **P@1**-val↑ | | **P@1**-test↑ | |
|---|---|---|---|---|
| **VLM** | VLM only | + LLM zero-shot | VLM only | + LLM zero-shot |
| `GD` | 60.09 | 58.05 | 59.32 | 58.47 |
| `GDrec` | 67.61 | 67.48 | 68.37 | 68.17 |
| `GD-1.5` | 47.67[†] | 59.00[†] | — | — |
| `Flo2` | 67.91 | 67.69 | 66.16 | 66.44 |

Table 7: **Results of wrapping the VLM's outputs with a zero-shot LLM.** 'VLM only' are the scores of the VLM without any adaptation, and '+ LLM zero-shot' corresponds to `LLM-wrapper` without any fine-tuning of the LLM (Llama 3 8B here). [†] see Table 5.

Figure 5: **Performance (P@1) of `LLM-wrapper` on RefCOCOg (val) with respect to the number of training samples.** We fine-tune Llama 3 8B on RefCOCOg (train).

## 5 CONCLUSION

This work introduces `LLM-wrapper`, a simple approach for the black-box adaptation of VLMs to the REC task, that leverages an LLM to reason on VLMs' outputs, translated into natural language. We demonstrate that `LLM-wrapper` significantly boosts the performance of VLMs, for several combinations of LLMs and VLMs, and, in some settings, even bridges the gap to classic fine-tuning. We also show how `LLM-wrapper` can ensemble predictions from different VLMs to leverage their respective strengths and how it can transfer across VLMs and to out-of-domain datasets. Thanks to efficient and well-studied methods for LLM fine-tuning, `LLM-wrapper` is simple to use in practice, requiring limited hyper-parameter tuning, and computationally efficient. Future works include relying on fewer examples to fine-tune `LLM-wrapper`, and applying `LLM-wrapper` to different tasks, such as text-video retrieval (Fang et al., 2021).

**Limitations.** `LLM-wrapper` comes with some limitations. First, an additional inference cost is introduced by integrating LLM reasoning on VLM outputs. Second, the effectiveness of `LLM-wrapper` relies on the quality of the underlying VLM since diverse and accurate bounding boxes are crucial for success. Third, bounding box information alone (box coordinates and labels) may not be sufficient for `LLM-wrapper` to 'understand' the scene in some cases. Examples are shown in Appendix E.2. To address this issue, a promising direction is to enhance `LLM-wrapper` by using the LLM to identify missing visual information and suggest augmented queries for the VLM. This would help disambiguate cases where additional visual cues are needed.

## ACKNOWLEDGEMENTS

We thank Pr. Eric Gaussier for his support and insightful feedbacks on this project. This work was performed using HPC resources from GENCI–IDRIS (Grant 2024-AD011014446) and with the support of the ANR MultiTrans (ANR-21-CE23-0032).

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

## A ADDITIONAL RESULTS: LLM-WRAPPER ON ALREADY REC-TUNED VLMS

We explore the use of `LLM-wrapper` on VLMs that are already optimized to deal with the REC task. This supplementary experiment aims to confirm whether `LLM-wrapper` is compatible with pre-existing REC-specific adaptations. Our analysis includes VLMs from our main experiments after white-box fine-tuning on REC data i.e., GD SwinB, GDrec SwinB and Flo2 fine-tuned. We also include an additional VLM, Kosmos-2 (Peng et al., 2023), that is directly designed to ground referring expressions. For each candidate VLM, we compare P@1 scores, with and without additional `LLM-wrapper` adaptation using RefCOCOg-train (umd) data. The evaluation is made on RefCOCOg val-umd and test-umd and results are shown in Table 8.

Table 8: **Results of `LLM-wrapper` on the REC task, when applied to already REC-adapted VLMs**, in P@1↑. 'FT' stands for 'fine-tuning' and '(B)' for the 'SwinB' backbone. [†] marks results directly taken from Contrastive Region Guidance (CRG) paper (Wan et al., 2024).

| | | | RefCOCOg | |
|---|---|---|---|---|
| **Adaptation** | **VLM** | **LLM** | val-umd | test-umd |
| White-box FT | GD(B) | | N/A[†] | 66.30[†] |
| White-box FT + CRG (Wan et al., 2024) | GD(B) | | N/A[†] | 69.60[†] ↑**3.30** |
| White-box FT | GD(B) | | 78.51 | 77.99 |
| White-box FT + LLM-wrapper | GD(B) | Mixtral | 82.31 ↑**3.80** | 82.15 ↑**4.16** |
| White-box FT + LLM-wrapper | GD(B) | Llama3 | 82.76 ↑**4.25** | 82.61 ↑**4.62** |
| White-box FT | GDrec(B) | | 80.19 | 79.85 |
| White-box FT + LLM-wrapper | GDrec(B) | Mixtral | 82.58 ↑**2.39** | 81.95 ↑**2.10** |
| White-box FT + LLM-wrapper | GDrec(B) | Llama3 | 81.66 ↑**1.47** | 81.47 ↑**1.62** |
| White-box FT | Flo2 FT | | 90.32 | 91.02 |
| White-box FT + LLM-wrapper | Flo2 FT | Mixtral | 90.40 ↑**0.08** | 90.92 ↓**0.10** |
| White-box FT + LLM-wrapper | Flo2 FT | Llama3 | 90.50 ↑**0.18** | 91.03 ↓**0.01** |
| Designed for REC | Kosmos-2 | | 60.60 | 61.41 |
| Designed for REC + LLM-wrapper | Kosmos-2 | Mixtral | 62.03 ↑**1.43** | 62.59 ↑**1.18** |
| Designed for REC + LLM-wrapper | Kosmos-2 | Llama3 | 62.09 ↑**1.49** | 62.39 ↑**0.98** |

While the performance gains are modest — an expected outcome as the VLMs are already adapted and optimized for the REC task —, it is important to note that `LLM-wrapper` avoids degrading performances by any significant value and, in some cases, achieves a slight improvement of up to +4.6 P@1. These results highlight the compatibility of `LLM-wrapper` with any state-of-the-art VLM, whether previously fine-tuned on REC data or not.

Evaluating `LLM-wrapper`, when applied to GD SwinB, also allows for direct comparison with related work Contrastive Region Guidance (CRG) (Wan et al., 2024), mentioned in Section 2. As shown in the first 5 rows of Table 8, `LLM-wrapper` displays both higher final scores and bigger score boosts than CRG on RefCOCOg test set, while remaining fully black-box and requiring a single VLM inference per input (text query, image) pair.

## B EVALUATING LLM-WRAPPER ON REFERRING EXPRESSION SEGMENTATION

We explore the use of `LLM-wrapper` on Referring Expression Segmentation (RES). This task, closely related to REC, consists in outputting a segmentation mask — instead of a bounding box — based on a (text query, image) input pair. Following Ren et al. (2024b), we use Segment Anything (SAM) (Kirillov et al., 2023) to convert predicted bounding boxes into segmentation masks. As for metrics, we follow Kazemzadeh et al. (2014); Mao et al. (2016); Lai et al. (2024) and compute cIoU (cumulative intersection over cumulative union on all referring expressions) and gIoU (average of referring expression-based IoUs, which is less biased in favor of large objects). This experiment aims to extend `LLM-wrapper` to new visual tasks, beyond REC. Moreover, the use of IoU-based metrics (instead of P@1 in REC) further tests the robustness of our predicted outputs.

Table 9: **Results of `LLM-wrapper` on the Referring Expression Segmentation (RES) task**, in cIoU/gIoU↑, alongside SOTA RES-specific models. 'FT' stands for 'fine-tuning', '(T)' and '(B)' for the 'SwinT' and 'SwinB' backbones. [†] indicates results directly taken from cited papers. Finally, 'SAM' indicates the use of Segment Anything (Kirillov et al., 2023) to convert bounding boxes into segmentation masks.

| | | | RefCOCOg (umd) | | | |
| | | | cIoU | | gIoU | |
| Method | VLM | LLM | val | test | val | test |
|---|---|---|---|---|---|---|
| LISA-7B FT (Lai et al., 2024) | | | 67.9[†] | 70.6[†] | N/A | N/A |
| GLaMM (Rasheed et al., 2024) | | | 74.2[†] | 74.9[†] | N/A | N/A |
| Zero-shot VLM + SAM | `GD(T)` | | 40.90 | 41.37 | 50.56 | 50.46 |
| Zero-shot VLM + `LLM-wrapper` + SAM | `GD(T)` | Mixtral | 57.80 ↑**16.9** | 58.00 ↑**16.6** | 64.71↑**14.2** | 65.01↑**14.6** |
| Zero-shot VLM + `LLM-wrapper` + SAM | `GD(T)` | Llama3 | 57.86 ↑**17.0** | 58.59 ↑**17.2** | 65.12↑**14.6** | 65.42↑**15.0** |
| White-box FT + SAM | `GD(B)` | | 57.93 | 58.53 | 65.62 | 65.96 |
| White-box FT + `LLM-wrapper` + SAM | `GD(B)` | Mixtral | 62.42 ↑**4.5** | 63.39 ↑**4.9** | 68.54 ↑**2.9** | 69.04↑**3.1** |
| White-box FT + `LLM-wrapper` + SAM | `GD(B)` | Llama3 | 62.44 ↑**4.5** | 63.84 ↑**5.3** | 68.81↑**3.2** | 69.41↑**3.5** |
| Zero-shot VLM + SAM | `Flo2` | | 48.12 | 47.04 | 57.24 | 56.19 |
| Zero-shot VLM + `LLM-wrapper` + SAM | `Flo2` | Mixtral | 59.38 ↑**11.3** | 58.49↑**11.5** | 66.17↑**8.9** | 65.59↑**9.4** |
| Zero-shot VLM + `LLM-wrapper` + SAM | `Flo2` | Llama3 | 59.53↑**11.4** | 59.58 ↑**12.5** | 66.07↑**8.8** | 66.00↑**9.8** |
| White-box FT + SAM | `Flo2 FT` | | 70.73 | 72.61 | 74.81 | 75.93 |
| White-box FT + `LLM-wrapper` + SAM | `Flo2 FT` | Mixtral | 70.81 ↑**0.1** | 72.50↓**0.1** | 74.87↑**0.1** | 75.89↓**0.04** |
| White-box FT + `LLM-wrapper` + SAM | `Flo2 FT` | Llama3 | 70.92 ↑**0.2** | 72.61 | 74.96↑**0.2** | 75.94↑**0.01** |

In Table 9, we show results of `LLM-wrapper` and of SOTA RES-specific methods — pixel grounding Large Multimodal Models (LMMs). The latter approach integrates image encoding information into LMMs, using vision-language projections and vocabulary augmentation with specialized tokens related to segmentation masks (Lai et al., 2024) or image region encodings (Rasheed et al., 2024). They are designed and trained, in a white-box end-to-end manner, to output fine-grained segmentation masks. Contrasting with this approach, `LLM-wrapper` is a model-agnostic black-box adaptation framework that emphasizes object-level reasoning. Table 9 shows that `LLM-wrapper` improves cIoU/gIoU scores in almost all cases, in accordance with our REC results in Table 3 and Table 8. Note that `LLM-wrapper`'s RES scores depend on the VLM being adapted and on SAM's box-to-mask conversion performance.

## C  ADDITIONAL ABLATIONS ON POSSIBLE VARIABLES IMPACTING PERFORMANCES

### C.1  IMPACT OF THE NUMBER OF FINE-TUNED PARAMETERS ON PERFORMANCE

Our ablation study on LoRA's rank $r$, presented in Figure 4b, shows the P@1 variations for different $r$ values. As the number of trainable parameters scales linearly with LoRA's rank $r$, this analysis highlights how trainable parameters count impacts performance. To ablate more precisely this aspect, we analyze the impact of the number of fine-tuned parameters on the P@1 performance in Table 10. In this table, we also report the original performance of the respective LLMs on the standard MMLU benchmark (Hendrycks et al., 2021) and HellaSwag (Zellers et al., 2019), a benchmark for LLM commonsense reasoning evaluation. As part of our analysis, we also compute the Pearson correlation between the P@1 scores and other variables from Table 10 and summarize results in Table 11.

Table 11 shows that the absolute number of fine-tuned parameters has only a loose correlation with performance (pearson=-0.07). This supports findings from Table 10 showing the limited impact of the number of fine-tuned parameters, such as:

- Variations in LoRA's rank $r$ do not have a large effect on the P@1 score of Llama3 8B on RefCOCOg (cf Table 10, rows 1 to 4).

- Llama3 8B ($r = 64$, 176M trainable parameters) and Mixtral 8×7B ($r = 128$, 114M trainable parameters) yield very similar P@1 scores (77.72 vs. 77.57), despite Llama3 8B fine-tuning a much higher percentage (2.15% vs. 0.24%) of parameters. The similarity of results, despite

Table 10: **Impact of the number of trainable parameters** on the performances of `GD` adapted by `LLM-wrapper` and evaluated on RefCOCOg.

| LLM | Rank ($r$) | Trainable params | Total params | Trainable params % | P@1 (val splits) | LLM MMLU | LLM HellaSwag |
|---|---|---|---|---|---|---|---|
| Llama3 8B | 12 | 33M | 8.1B | 0.41 % | 77.84 | 66.6 | 82 |
| Llama3 8B | 64 | 176M | 8.2B | 2.15% | 77.72 | 66.6 | 82 |
| Llama3 8B | 128 | 352M | 8.4B | 4.20 % | 78.12 | 66.6 | 82 |
| Llama3 8B | 192 | 529M | 8.6B | 6.18% | 77.70 | 66.6 | 82 |
| Mixtral 8x7B | 128 | 114M | 46.8B | 0.24 % | 77.57 | 70.6 | 84.4 |
| Gemma-2 9B | 128 | 465M | 9.7B | 4.79 % | 74.12 | 71.3 | 81.9 |
| Gemma-2 2B | 128 | 199M | 2.8B | 7.08 % | 70.51 | 52.2 | 72.9 |

Table 11: **Pearson Correlation with `LLM-wrapper`'s P@1** for various variables.

| Variable | Pearson Correlation with P@1 |
|---|---|
| LLM Total Params (count) | 0.33 |
| LLM Trainable Params (count) | -0.07 |
| LLM Trainable Params (%) | -0.64 |
| LLM MMLU score | 0.72 |
| LLM HellaSwag score | 0.88 |

different fine-tuning settings, could then be due to architectural differences, with Mixtral being a Mixture of Experts model.

- Gemma-2 9B ($r = 128$, 465M trainable params) underperforms compared to Llama3 8B ($r = 64$, 176M trainable params), even though its setting includes fine-tuning more than double the number of parameters.

These findings suggest that the architecture of the LLM and training specifics may have more influence on performances. Furthermore, our analysis finds a strong correlation between `LLM-wrapper`'s P@1 scores and the LLM's performances on reasoning tasks, with a Pearson correlation of 0.88 with HellaSwag. In conclusion, the number of fine-tuned parameters has a small impact, but the LLM's architecture and original performance on reasoning tasks play a more significant role on `LLM-wrapper`'s REC performance.

## C.2 IMPACT OF INPUT COMPLEXITY ON `LLM-WRAPPER`'S PERFORMANCE

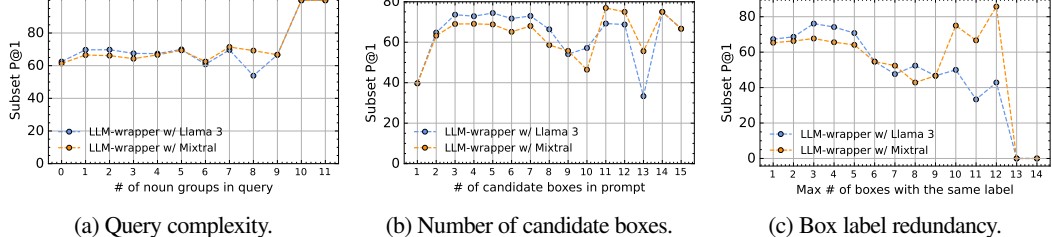

(a) Query complexity.  (b) Number of candidate boxes.  (c) Box label redundancy.

Figure 6: **Study of `LLM-wrapper`'s sensitivity to input complexity.** `LLM-wrapper`'s performance in P@1 on subsets of the data with varying levels of complexity, coming from the query (number of noun groups in the query) or the box listing in the prompt (number of candidate boxes and box label redundancy, defined as the highest number of boxes with the same label detected for a query). `LLM-wrapper` is used with `Flo2` and subsets are aggregated from RefCOCO/+/g and Talk2Car val sets.

In Figure 6, we aggregate results from RefCOCO/+/g and Talk2Car val sets to show `LLM-wrapper`'s P@1 scores on different data subsets. These subsets are split based on varying levels of input complexity, with respect to the number of noun groups in the query (Fig. 6a), the number of candidate boxes in the prompt (Fig. 6b) and the redundancy of box labels (Fig. 6c). We first study in Figure 6a, `LLM-wrapper`'s

P@1 performances when the the number of noun groups[1] increases. For both Llama 3 8B and Mixtral, we see that `LLM-wrapper` is robust to, and even benefits from, increasing textual complexity, with stable scores around 67 P@1 for 0 to 9 noun groups, and 100 P@1 for 10 to 11 noun groups. We then display in Figure 6b how performance evolves with respect to the number of boxes listed in the prompt. `LLM-wrapper` is robust to an increasing number of candidates, with a P@1 around 66 for 2 to 15 boxes, and only a few performance variations for more than 8 boxes. Finally, we test a finer-grained notion of box label redundancy, defined as the highest number of boxes with the same label detected for a query. Figure 6c shows how performance changes based on this redundancy. For both Llama 3 8B and Mixtral, a clear decreasing tendency is observed on the performance as more boxes share a same label, in particular when that is the case for more than 6 boxes. This study indicates that `LLM-wrapper` is robust to increasing levels of input query and box listing complexity, as long as the detected boxes display diversified labels.

## D  FAILURE CASE ANALYSIS

A failure case arises when the LLM does not succeed in producing the index of a candidate bounding box as output. This happens if the LLM outputs a non-integer value or an integer that is out of range with respect to the candidate boxes' list. However, these issues are very rare. In our experiments, a fine-tuned `LLM-wrapper` shows 0% of non-integer output instances, and only the issue of out of range integer generation remains. In Table 12, we report the percentage of instances where the output integer is out of range, when `Flo2` is adapted with `LLM-wrapper`, using Mixtral or Llama 3 8B. We observe that such issues occur in less than $0.3\%$ of instances. In these rare cases, we use a simple fallback strategy: the best-ranked box from the zero-shot VLM is used as prediction. For qualitative examples of failure cases due to reasoning issues, rather than generation issues, intuitions and visualizations are given in Appendix E.2.

Table 12: **Percentages of out of range integer generation** when using `LLM-wrapper` on `Flo2`. [†] indicates that results are obtained while using a transferred `LLM-wrapper`, trained on RefCOCOg.

| Method | LLM | RefCOCOg val | RefCOCOg test | RefCOCO val | RefCOCO test | RefCOCO+ val | RefCOCO+ test | Talk2Car val | Talk2Car test | HC-RefLoCo val | HC-RefLoCo test |
|---|---|---|---|---|---|---|---|---|---|---|---|
| LLM-wrapper | Mixtral | 0.04% | 0.03% | 0.01% | 0% | 0.02% | 0.03% | 0% | 0% | — | — |
| LLM-wrapper | Llama3 | 0% | 0% | 0.30% | 0.22% | 0.06% | 0.09% | 0% | 0% | 0%[†] | 0%[†] |

## E  ADDITIONAL QUALITATIVE EXAMPLES

In this section, we use `LLM-wrapper` with `Flo2` and a Llama3 8B fine-tuned on RefCOCOg data.

### E.1  QUALITATIVE SUCCESS CASES ON COMPLEX QUERIES

We show qualitative examples of `LLM-wrapper`'s successes against white-box fine-tuned `Flo2` on HC-RefLoCo in Figure 7 and Figure 8. In these examples, `LLM-wrapper` is able to properly process more than 10 candidate boxes, aligned with long and complex queries, to identify the correct box, while white-box fine-tuned `Flo2` is predicting distractor objects.

### E.2  QUALITATIVE FAILURE CASES

We display qualitative examples of `LLM-wrapper`'s failures against zero-shot `Flo2` on RefCOCOg. In particular, as mentioned in Section 5, the main failure case of `LLM-wrapper` occurs when bounding box coordinates and labels alone are insufficient to ground certain expressions that require additional visual cues. We give various examples showing this type of failure in Figure 9. For instance, `LLM-wrapper` is missing visual information on the zebra's head direction in Fig. 9a, on the relative position of the curtains and chairs in Fig. 9b and on the position of each person with respect to the camera in Fig. 9c.

---

[1]Noun groups in text queries are identified using Spacy's *noun_chunks* method (Honnibal et al., 2020).

Query: *"This individual appears to be a woman with long, straight blonde hair that drapes over her right shoulder. She is wearing a light-colored, possibly cream or pale yellow blazer. The woman is engaged in an activity where she is bringing a clear glass, which she holds in her right hand, towards her mouth as if to take a sip of a beverage. Her left hand is not visible in the image. She is positioned on the far left in the group of people captured in the photograph."*

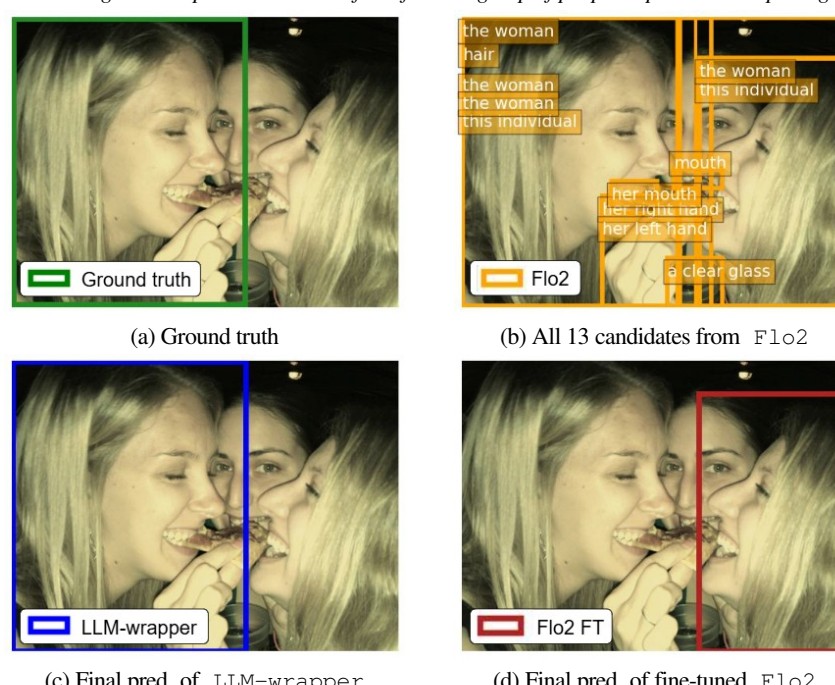

(a) Ground truth        (b) All 13 candidates from `Flo2`

(c) Final pred. of `LLM-wrapper`      (d) Final pred. of fine-tuned `Flo2`

Figure 7: **First qualitative result of `LLM-wrapper` tuned on RefCOCOg, evaluated on HC-RefLoCo.** `LLM-wrapper` takes multiple candidates from zero-shot `Flo2` as inputs (in orange in Fig. 7b) to identify the best box (in **blue**, in Fig. 7c), while white-box fine-tuned `Flo2` fails (in **red** in Fig. 7d).

Query: *"The individual is a middle-aged man with short, dark hair, appearing startled or comically alarmed. He is wearing a pale dress shirt and is positioned as if emerging from a mirror, with his left side showing."*

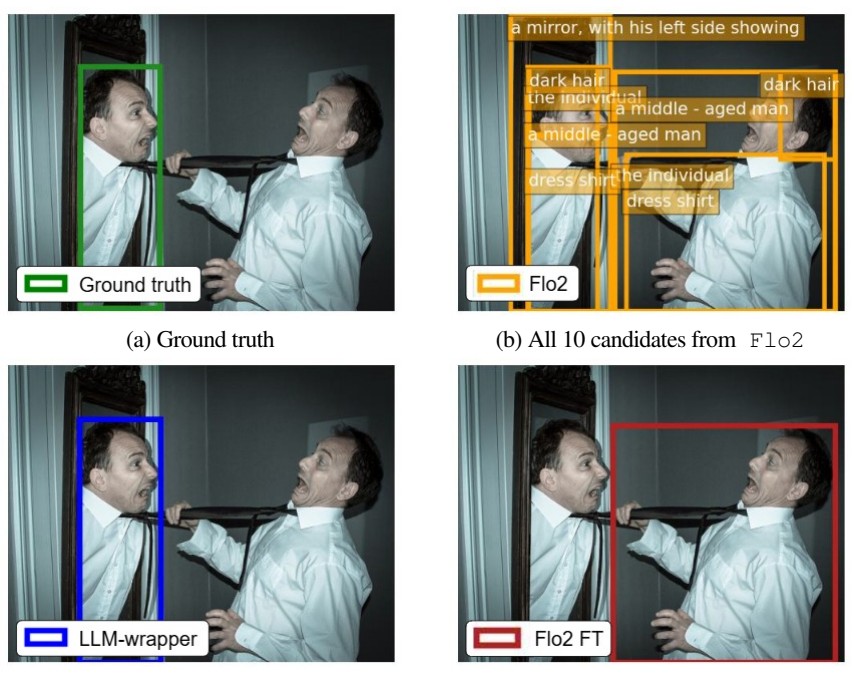

(a) Ground truth        (b) All 10 candidates from `Flo2`

(c) Final pred. of `LLM-wrapper`      (d) Final pred. of fine-tuned `Flo2`

Figure 8: **Second result of `LLM-wrapper` evaluated on HC-RefLoCo.** Same legend as Figure 7.

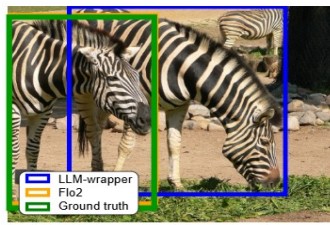 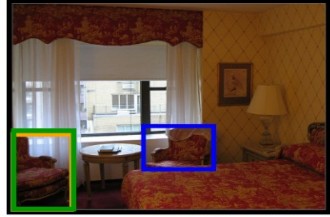 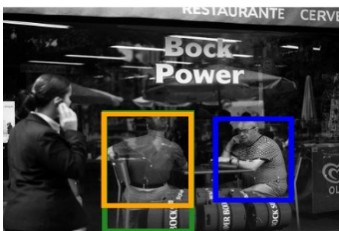

(a) Query: *"Zebra whose head is not facing down."*     (b) Query: *"Chair without the curtain on it"*     (c) Query: *"A man with his back turned toward the camera, enjoying a conversation with a friend."*

Figure 9: **Visualizations on RefCOCOg of `LLM-wrapper`'s main failure case.** `LLM-wrapper`'s predictions are in **blue** vs. zero-shot `Flo2`'s predictions in **orange**. In these three examples, `LLM-wrapper` fails to identify the best candidate box as coordinates and labels are not providing enough visual cues for the LLM to choose correctly. For instance, `LLM-wrapper` is missing visual information on the zebra's head direction (Fig. 9a), on the relative position of the curtains and chairs (Fig. 9b) and on the position of each person with respect to the camera (Fig. 9c).

Going beyond `LLM-wrapper`'s main failure scenario, we add visualizations of corner failure cases in Figure 10. They illustrate possible issues hindering the LLM's reasoning, i.e., when no proper candidate box is identified by the zero-shot VLM in the first place (Fig. 10a), when the VLM fails to detect important contextual objects, necessary to 'perceive' the scene and localize the object of interest (Fig. 10b), when rich candidate boxes exist but without adequate labels (Fig. 10c).

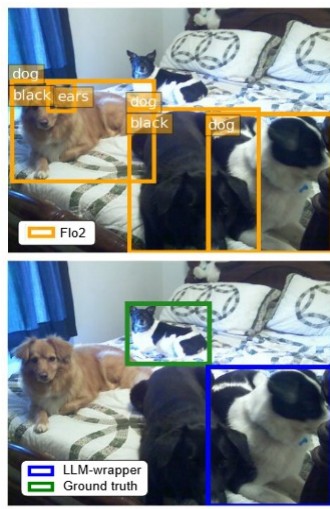 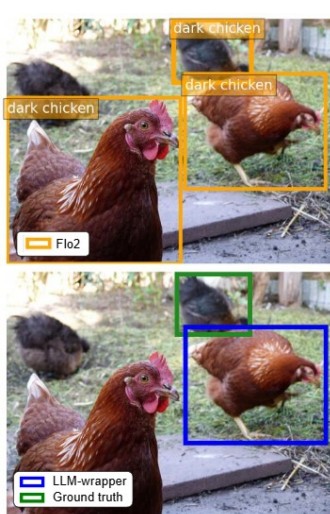 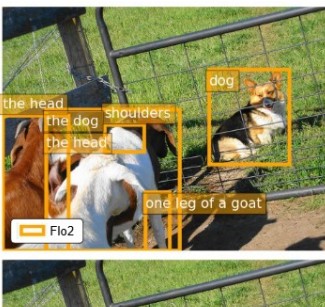
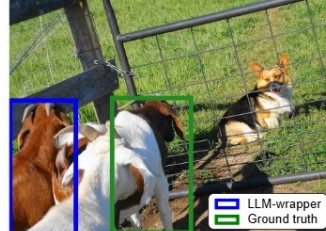

(a) Query: '*Black and white dog with pointy ears.*'     (b) Query: '*Dark chicken closest to the fence.*'     (c) Query: '*The head and shoulders and one leg of a goat closest to the dog.*'

Figure 10: **Visualizations of three corner failure cases of `LLM-wrapper` on RefCOCOg.** As we use `LLM-wrapper` to adapt a zero-shot `Flo2`, we show on the first row in **orange**, for each sample, the respective `Flo2`'s candidate boxes that `LLM-wrapper` takes as inputs. In the second row, we visualize the prediction of `LLM-wrapper` in **blue**, chosen among `Flo2`'s candidates, as well as the ground truth box. We observe that the LLM reasoning can be hindered if candidate boxes are missing the object of interest (the correct dog in Fig. 10a), necessary contextual objects (the fence in Fig. 10b) or proper labels (goats are detected but not properly labeled in Fig. 10c, bringing confusion with respect to the scene description).

