# OpenReview forum: "LLM-wrapper: Black-Box Semantic-Aware Adaptation of Vision-Language Models for Referring Expression Comprehension"
_ICLR.cc/2025/Conference — ICLR 2025 Poster_

### Official Review · Reviewer_o1yY · 2024-10-27

**Soundness:** 3
**Presentation:** 3
**Contribution:** 3
**Rating:** 6
**Confidence:** 4

**Summary:**

This paper proposes a method called LLM-wrapper, aiming at ‘black-box’ adaptation of VLMs for the REC task using Large Language Models.

**Strengths:**

1. The idea is impressive.
2. The paper is a well-structured format.

**Weaknesses:**

1. The performances of this method is limited. The performances of many REC methods such as [1], [2], [3] don’t rely on the LLM are better than the performances of this method.
2. The experimental evaluation of this method is constrained. Additional dataset comparisons, such as Flickr30K entities and RefClef, should be taken into account.

[1] Qiu, Heqian and Li, Hongliang and Zhao, Taijin and Wang, Lanxiao and Wu, Qingbo and Meng, Fanman, RefCrowd: Grounding the Target in Crowd with Referring Expressions, MM2022.

[2] Zhao, Peizhi and Zheng, Shiyi and Zhao, Wenye and Xu, Dongsheng and Li, Pijian and Cai, Yi and Huang, Qingbao, Rethinking Two-Stage Referring Expression Comprehension: A Novel Grounding and Segmentation Method Modulated by Point, AAAI2024

[3] Jing, Chenchen and Wu, Yuwei and Pei, Mingtao and Hu, Yao and Jia, Yunde and Wu, Qi, Visual-Semantic Graph Matching for Visual Grounding, MM2020.

**Questions:**

please refer to above-mentioned.

---

> ### Author Response · Authors · 2024-11-21
> **Answer to Reviewer o1yY**
>
> We thank the reviewer for the feedback and comments.
>
> **R-o1yY-W1:** *REC performances.*
>
>
> **R-o1yY-A1:** Indeed, the performances of many REC methods are better than the performances of LLM-wrapper, and we thank the reviewer for the references [1],[2],[3] that we will add to the paper.
>
> However, this observation misses the key focus of our work: the aim of LLM-wrapper is not to replace or outperform traditional white-box learning and fine-tuning methods for the REC task, as clarified in the paper (Lines 27, 64, 88, 352).
> Instead, our method is designed as an alternative solution for scenarios where traditional white-box learning and fine-tuning approaches are not feasible, which includes:
> * When there is no access to a model's architecture or weights, such as in the case of proprietary or closed-source models (Lines 17, 50–52).
> * When expertise in designing fine-tuning objectives and optimizing hyperparameters specific to each VLM is lacking (Lines 48–49).
> * When using fine-tuning APIs is not an option, for instance, due to concerns about sharing private data (Lines 52–53).
>
> Additionally, LLM-wrapper provides unique advantages over classic white-box fine-tuning:
> * *Transferability*: Once fine-tuned on one VLM, LLM-wrapper can be directly transferred to another VLM without additional adaptation.
> * *Ensembling capability*: LLM-wrapper can seamlessly learn to combine outputs from multiple VLMs, leveraging their strengths for improved performance.
>
>
> **R-o1yY-W2:** *Additional dataset comparisons.*
>
>
> **R-o1yY-A2:** Thank you for the suggestion to expand the experimental evaluation.
>
> In response, we evaluate LLM-wrapper on the HC-RefLoCo dataset, which features longer and more complex text queries than RefCOCO/+/g, with 84.4 words per query on average. Since HC-RefLoCo lacks a training set, we transfer an LLM-wrapper trained on RefCOCOg using the protocol described in Section 4.3 (starting L.408).
>
>
> LLM-wrapper achieves  a significant +18.8 (resp. +18.4) P@1 improvement on the val (resp. test) split over zero-shot Florence-2 (see _Table R.11_ below), showing its ability to handle complex queries, adapt to more challenging datasets, and reaffirming its transferability. We will add these results to the revised paper.
>
> | Method      | VLM       / LLM     | Training Data  | P@1 (HC-RefLoCo val) | P@1 (HC-RefLoCo test) |
> | ----------- | ------------------- | -------------- | -------------------- | --------------------- |
> | 0-shot VLM  | Florence-2          | None           | 48.64                | 48.87                 |
> | LLM-wrapper | Florence-2 / Llama3 | RefCOCOg train | 67.40                | 67.26                 |
>
> _Table R.11. LLM-wrapper performance on the HC-RefLoCo benchmark (when fine-tuning on RefCOCOg), vs zero-shot Florence 2._
>
> Regarding RefClef, we note that 99.7% of its queries have fewer than three words, with 76.5% consisting of just a single word. These short queries are less aligned with LLM-wrapper’s strengths in reasoning over complex expressions involving relations, spatial cues, or negations. Thus, we do not anticipate substantial improvements on RefClef. For Flickr30K Entities, the black-box adaptation with LLM-wrapper is ongoing. We aim to include those results in the revised paper if time allows, or in the final version.

---

> ### Author Response · Authors · 2024-11-26
> **Feedbacks were incorporated in the revised paper**
>
> Dear Reviewer,
>
> As promised, we have incorporated your feedback and added clarifications and results to the revised paper, where changes are marked in blue. Below, we point to the specific sections where the changes addressing your remarks and questions have been implemented.
>
> **R-o1yY-A2:** Our answer has been integrated to the revised paper, with the new Table 6 and new paragraph at the end of Section 4.3 (Paragraph: "Transferring a trained LLM-wrapper to new datasets.").
>
> As the author-reviewer discussion period continues, we would greatly value your feedback on whether our responses have sufficiently addressed your concerns.
> If you have any further comments, please feel free to share them with us.
>
> Best regards,

---

> ### Comment · Reviewer_o1yY · 2024-11-26
>
> . This method could be a viable option when the weights or architecture of LLMs are not available. However, existing LLMs such as GPT-4, Claude, OFA, and MDETR have demonstrated their ability to handle various vision-language tasks, including REC. Moreover, some of these models have publicly released their code. In extreme cases where none of these LLMs are effective, could you provide a specific example and present experimental results for such scenarios? Based on the authors' response, I decide to raise the rating to 5.

---

> > ### Author Response · Authors · 2024-11-28
> > **Answer part 1/2**
> >
> > (Answer part 1/2)
> >
> > Thank you for your message.
> >
> > Indeed, some LLMs like GPT-4V, can now handle both text and images and we classify these models as VLMs. These models are not in competition with our method but rather can be enhanced for REC by our black-box adaptation, either instead of, or alongside, traditional fine-tuning.
> >
> > In particular, OFA is similar to Florence-2, a sequence-to-sequence model with multitask learning, while MDETR and Grounding-DINO are both DETR-like models. In the paper, we applied LLM-wrapper to Florence-2 and Grounding-DINO because they perform better on RefCOCO/+/g than OFA and MDETR. We did not use models like GPT-4V or Claude in our main experiments due to the cost of acquiring training data via their API. However, to avoid the API cost, we demonstrated that transferring the LLM-wrapper adaptation from one VLM to another is effective, as shown with Grounding-DINO 1.5's free API calls (section 4.3 and table 5). Our experiments show that LLM-wrapper is model-agnostic and can be applied to any VLM, open-source or behind API, as long as it processes vision-language pairs and outputs bounding boxes.
> >
> > Regarding case scenarios where LLM-wrapper is more effective, we show in the updated section 4.3 that LLM-wrapper can outperform traditional fine-tuning for adapting VLMs. Indeed, when applied in a zero-shot manner to datasets with unseen textual complexity like HC-RefLoCo, VLMs adapted with LLM-wrapper perform better than those using classic white-box fine-tuning. To show that, we compare in _Table R.12_ below different adaptation methods on Florence-2, the SOTA VLM on standard REC benchmarks (RefCOCO/+/g).
> >
> >
> > | Adaptation                          | VLM                                                        | Fine-tuning REC Data                    | P@1 (val+test)                                    |
> > | ------------------------------------------ | ---------------------------------------------------------- | --------------------------------------- | ------------------------------------------------- |
> > | None (0-shot VLM)                          | Florence-2                                                 | None                                    | 48.8                                              |
> > | Classic fine-tuning (white-box adaptation) | Florence-2                                                 | Multiple datasets including RefCOCO/+/g | 56.0 **(↑ 7.2)** |
> > | LLM-wrapper (black-box adaptation)         | Florence-2 (grounding boxes)                               | RefCOCOg                                | 67.3 **(↑18.5)** |
> > | LLM-wrapper (black-box adaptation)         | Florence-2 (ensembling grounding and region caption boxes) | RefCOCOg                                | 71.7 **(↑22.9)** |
> >
> > _Table R.12. Evaluation on HC-RefLoCo of Florence-2 adapted with LLM-wrapper, vs zero-shot Florence-2 and white-box fine-tuned Florence-2. (LLM-wrapper is implemented with Llama3 8B)._
> >
> > *Table R.12* shows that the LLM-wrapper adaptation of Florence-2 using RefCOCOg data, significantly outperforms zero-shot Florence 2 (**↑18.5**) and provides a larger performance boost than classic white-box fine-tuning on RefCOCO/+/g (**↑7.2**). The performance boost of LLM-wrapper can be further improved by using its ensembling capabilities (**↑22.9**, last row). This suggests that LLM-wrapper adapts VLMs with more robust, general knowledge of object localization, while white-box fine-tuning tends to overfit to specific types of REC data, like the short text queries in RefCOCO/+/g.
> >
> > The results are not only robust but also competitive, when compared to OFA and GPT-4V, as shown in *Table R.13*:
> >
> > | Method                        | P@1 (val+test) |
> > | ----------------------------- | -------------- |
> > | GPT-4V #                      | 17.4           |
> > | OFA #                         | 48.4           |
> > | OFA-Large #                   | 70.5           |
> > | Flo2 adapted with LLM-wrapper | 71.7           |
> >
> > _Table R.13. Florence 2 adapted with LLM-wrapper vs OFA and GPT-4V on the HC-RefLoCo benchmark. (# indicates results coming from the HC-RefLoCo paper [1])._
> >
> > (Qualitative results of Florence-2 adapted with LLM-wrapper and evaluated on HC-RefLoCo, are shown in Appendix D.1 of the revised paper.)

---

> > > ### Comment · Reviewer_o1yY · 2024-12-03
> > >
> > > Thanks for the replies. This replies address most of my concerns. So, I  decide to rise the rate to 6.

---

> > ### Author Response · Authors · 2024-11-28
> > **Answer part 2/2**
> >
> > (Answer part 2/2)
> >
> >
> > Regarding open-source models that are already very capable on REC, we included to the revised paper (Appendix A) results of LLM-wrapper applied to such models. While we did not expect major improvements on classic REC data, our results show that *LLM-wrapper can still provide an additional boost, up to +2.4 in P@1*, even for models already adapted for REC through white-box fine-tuning.
> >
> >
> > In summary, LLM-wrapper is not intended to compete with the performance of specific VLMs. Instead, it is a lightweight and robust adaptation method that can improve the performance of all VLMs on the REC task, either as an alternative to, or alongside, other adaptation methods.
> >
> > Best regards,
> >
> > [1] F. Wei, J. Zhao, K. Yan, H. Zhang, C. Xu. A Large-Scale Human-Centric Benchmark for Referring Expression Comprehension in the LMM Era. NeurIPS 2024.

---

### Official Review · Reviewer_DAad · 2024-10-30

**Soundness:** 3
**Presentation:** 4
**Contribution:** 3
**Rating:** 8
**Confidence:** 5

**Summary:**

This paper introduces LLM-wrapper, a novel method for enhancing the Referring Expression Comprehension (REC) capabilities of black-box VLMs without directly fine-tuning them. LLM-wrapper leverages a fine-tuned LLM to refine the bounding box proposals generated by a frozen, zero-shot VLM. Specifically, LLM-wrapper takes in a natural language description of each proposal along with spatial coordinates and the original referring expression, then outputs the proposal most relevant to the given expression as the final REC prediction. This black-box adaptation approach allows closed-source VLMs to be adapted without access to their internal structure or gradients, making it adaptable to any VLM and capable of integrating proposals from multiple VLMs to improve performance through ensembling.

**Strengths:**

1. The paper is well-written, making the proposed method easy to understand with clear motivation and steps.

2. LLM-wrapper only needs to fine-tune the LLM once, allowing it to work with newer, more powerful VLMs without retraining, making it adaptable as VLM technology advances.

3. By combining outputs from multiple VLMs, LLM-wrapper reaches performance levels that compete with some fully fine-tuned VLMs, showing its robustness.

4. The method requires only a single GPU for fine-tuning, keeping training costs low and making it accessible.

5. Black-box approach is ideal for closed-source VLMs, adding value in scenarios where model internals aren’t accessible.

**Weaknesses:**

1. The LLM-wrapper approach doubles the inference cost compared to existing VLMs.

2. The effectiveness of the LLM-wrapper is highly dependent on the performance of the chosen VLM.

3. Several recent VLMs, such as Ferret [1], KOSMOS-2 [2], and Qwen-VL [3], support both referring expression comprehension (REC) and grounding. The paper could be strengthened by including more baseline experiments that compare the original REC performance of these VLMs with their LLM-wrapped versions.

4. In the LLM-wrapper paradigm, the number of entities in the referring expression likely influences the final performance. If the VLM can accurately predict all entities in a referring expression, further analysis is warranted on whether an increased number of entities aids or hinders the LLM's ability to ground the correct one. An ablation study examining the effect of entity count on performance would be valuable.

5. Extending point 4, the experiments are conducted on the RefCOCO series, which contains relatively simple referring expressions (with a length of 3.6–8.4 words, as reported in the paper). This limitation suggests a confined number of entities per sentence. It would be insightful to evaluate the model’s performance on longer expressions with more entities, such as HC-RefLoCo benchmark, to determine how the LLM performs when faced with more complex captions and a greater number of entities.

[1] H. You, H. Zhang, Z. Gan, X. Du, B. Zhang, Z. Wang, L. Cao, S.-F. Chang, and Y. Yang. Ferret: Refer
and ground anything anywhere at any granularity. arXiv preprint arXiv:2310.07704, 2023.

[2] Z. Peng, W. Wang, L. Dong, Y. Hao, S. Huang, S. Ma, and F. Wei. Kosmos-2: Grounding multimodal
large language models to the world. arXiv preprint arXiv:2306.14824, 2023.

[3] Bai, S. Bai, S. Yang, S. Wang, S. Tan, P. Wang, J. Lin, C. Zhou, and J. Zhou. Qwen-VL: A versatile vision-language model for understanding, localization, text reading, and beyond. arXiv preprint arXiv:2308.12966, 2023.

**Questions:**

Please refer to weaknesses 4 and 5.

---

> ### Author Response · Authors · 2024-11-21
> **Answer Part 1/2 to Reviewer DAad**
>
> **Answer Part 1/2**
>
> We thank the reviewer for the insightful feedback and comments.
>
> **R-DAad-W1:** *Double inference cost.*
>
>
> **R-DAad-A1:** We acknowledge this limitation of our approach. LLM-wrapper doubles the inference cost because it integrates LLM reasoning with the generalist VLM. However, our method is compatible with techniques that reduce LLM computation costs, such as FlashAttention and Quantization (L.204). We appreciate this comment and will include this limitation in the revised paper.
>
> **R-DAad-W2:** *LLM-wrapper is highly dependent on the performance of the chosen VLM.*
>
>
> **R-DAad-A2:** We agree that LLM-wrapper's performance depends on the quality of the chosen VLM. As acknowledged in the paper (L.42-43, L.159), LLM-wrapper leverages the ability of detection-oriented VLMs to localize most query nouns with reasonably accurate bounding boxes and labels. This assumption is reasonable given the significant improvements in generalist VLMs, which now perform well in grounding objects from text queries. This dependency is fundamental to our approach, and if a VLM fails to meet this assumption, LLM-wrapper's effectiveness would also diminish. We will further acknowledge this in the limitation section of the paper.
>
> **R-DAad-W3:** *Applying LLM-wrapper on other VLMs, like Ferret, KOSMOS-2, and Qwen-VL.*
>
>
> **R-DAad-A3:** We thank the reviewer for this suggestion. As the proposed VLMs (Ferret, Kosmos-2, Qwen-VL) are pre-trained for the REC task, we do not anticipate significant improvements with LLM-wrapper. Indeed, these VLMs are already designed to handle referring expressions and grounding tasks effectively. Nevertheless, we evaluate Kosmos-2 with and without LLM-wrapper on the RefCOCOg validation set. The LLM-wrapper here uses a Llama3 model trained to adapt Florence-2 (following our transfer protocol, similar to Section 4.3), providing a lower bound for what we could achieve with direct black-box fine-tuning of Kosmos-2 with LLM-wrapper, which is currently under investigation. While the improvement is small, it demonstrates that LLM-wrapper can slightly enhance the REC performance:
>
> | VLM      | Further adaptation                                  | P@1 (RefCOCOg val) |
> | -------- | --------------------------------------------------- | ------------------ |
> | Kosmos-2 | --                                                  | 60.60              |
> | Kosmos-2 | LLM-wrapper w/ Llama3 (transferred from Florence-2) | 61.29              |
>
> _Table R.7. Results when transferring LLM-wrapper: Florence-2  &rarr; Kosmos-2._
>
> To gather additional results regarding the use of LLM-wrapper on VLMs that are already REC-adapted, we add in _Table R.8_ below results of LLM-wrapper applied to Florence-2 Large FT and GDRec (SwinB), both of which are already fine-tuned on REC data. Further adapting them with LLM-wrapper produces the following results:
>
> | Method                                            | Further adaptation     | P@1 (RefCOCOg val) |
> | ------------------------------------------------- | ---------------------- | ------------------ |
> | Florence-2 Large FT after *white-box fine-tuning* | --                     | 90.32              |
> | Florence-2 Large FT after *white-box fine-tuning* | LLM-wrapper w/ Mixtral | 90.38              |
> | Florence-2 Large FT after *white-box fine-tuning* | LLM-wrapper w/ Llama3  | 90.50              |
> | GDRec (SwinB) after *white-box fine-tuning*       | --                     | 80.19              |
> | GDRec (SwinB) after *white-box fine-tuning*       | LLM-wrapper w/ Mixtral | 82.84              |
> | GDRec (SwinB) after *white-box fine-tuning*       | LLM-wrapper w/ Llama3  | 81.68              |
>
> _Table R.8. LLM-wrapper results when applied to VLMs, already fine-tuned for the REC task._
>
> While the improvements are modest, it is notable that LLM-wrapper does not degrade performance and, in some cases, provides a slight boost. This highlights its compatibility with state-of-the-art VLMs. We will include these results in the revised version of the paper.

---

> ### Author Response · Authors · 2024-11-21
> **Answer Part 2/2 to Reviewer DAad**
>
> **Answer Part 2/2**
>
>
> **R-DAad-W4:** *Analysis on the number of entities in the referring expression.*
>
>
> **R-DAad-A4:**
> We appreciate the reviewer's ablation study suggestion. To perform this study, we count the average number of words and noun groups per query for all of our datasets (in particular, for their val splits).
> Noun groups were identified using SpaCy’s `noun_chunks` method (e.g., the sentence *"Green plant behind a table visible behind a lady's head"* yields three noun groups: *"Green plant,"* *"a table,"* and *"a lady's head"*). The results are summarized in the _Table R.9_ below (note that we add numbers regarding the HC-RefLoCo dataset, with implementation details discussed in **R-DAad-A5**):
>
> | Dataset (val) | Avg. # of Words | Avg. # of Noun Groups | Avg. # of Boxes | P@1 (LLM-wrapper w/ Flo2, Llama3) | P@1 Boost (vs. Zero-shot VLM) |
> | ------------- | --------------- | --------------------- | --------------- | --------------------------------- | ----------------------------- |
> | RefCOCO       | 3.6             | 1.4                   | 2.8             | 70.19                             | +13.9                         |
> | RefCOCO+      | 3.6             | 1.6                   | 2.8             | 61.95                             | +8.2                          |
> | RefCOCOg      | 8.3             | 2.7                   | 3.7             | 77.94                             | +9.7                          |
> | HC-RefLoCo    | 84.4            | 22.9                  | 17.6            | 67.40                             | +18.8                         |
>
> _Table R.9. Text query complexity, number of candidate boxes and LLM-wrapper's P@1 on all our datasets (statistics and P@1 scores are computed on val splits)._
>
> LLM-wrapper demonstrates robust performance across datasets, showing no clear correlation between the number of entities in referring expressions and either the P@1 score or the performance boost. Interestingly, on the challenging HC-RefLoCo dataset, which features long queries with many entities (more than 20 noun groups per query on average), LLM-wrapper maintains strong reasoning capabilities and performs well on the numerous boxes proposed by the VLM (more than 17 per prompt on average).
> We will include this discussion in the revised paper.
>
>
> **R-DAad-W5:** *Evaluation of LLM-wrapper on longer expressions with more entities, such as HC-RefLoCo.*
>
>
> **R-DAad-A5:** At the reviewer’s suggestion, we evaluate LLM-wrapper on the HC-RefLoCo dataset, which includes longer expressions with more entities than the RefCOCO/+/g series. Since HC-RefLoCo has no training set, we *transfer* an LLM-wrapper trained on RefCOCOg to this dataset, with a similar protocol as described in Section 4.3 (starting L.408).
>
> LLM-wrapper achieves a significant +18.8 (resp. +18.4) P@1  improvement on the val (resp. test) split over zero-shot Florence-2 (see _Table R.10_ below), showing its ability to handle complex reasoning, adapt to datasets with more challenging expressions, and stressing again the transfer abilities of LLM-wrapper. These results will be added to the revised paper.
>
> | Method      | VLM       / LLM     | Training Data  | P@1 (HC-RefLoCo val) | P@1 (HC-RefLoCo test) |
> | ----------- | ------------------- | -------------- | -------------------- | --------------------- |
> | 0-shot VLM  | Florence-2          | None           | 48.64                | 48.87                 |
> | LLM-wrapper | Florence-2 / Llama3 | RefCOCOg train | 67.40                | 67.26                 |
>
> _Table R.10. LLM-wrapper performance on the HC-RefLoCo benchmark (when fine-tuning on RefCOCOg), vs zero-shot Florence 2._

---

> ### Author Response · Authors · 2024-11-26
> **Feedbacks were incorporated in the revised paper**
>
> Dear Reviewer,
>
> As promised, we have incorporated your feedback and added clarifications and results to the revised paper, where changes are marked in blue. Below, we point to the specific sections where the changes addressing your remarks and questions have been implemented.
>
> **R-DAad-A1** and **R-DAad-A2:** We added a "Limitations" paragraph in the Conclusion.
>
> **R-DAad-A3:** Our answer has been integrated to the revised paper in Section 4.2 and further expanded upon in Appendix A.
>
> **R-DAad-A4:** Our answer has been integrated to the revised paper in Section 4.3 and further expanded upon in Appendix B.2
>
> **R-DAad-A5:** Our answer has been integrated to the revised paper, with the new Table 6 and new paragraph at the end of Section 4.3 (Paragraph: "Transferring a trained LLM-wrapper to new datasets.").
>
> As the author-reviewer discussion period continues, we would greatly value your feedback on whether our responses have sufficiently addressed your concerns.
> If you have any further comments, please feel free to share them with us.
>
> Best regards,

---

> > ### Comment · Reviewer_DAad · 2024-12-02
> > **Thanks for the Rebuttal**
> >
> > Thank you for your comprehensive responses. The supplementary experiments have effectively addressed my concerns, leading me to raise my score. Additionally, I remain curious about the performance of the LLM-wrapper in dense entity environments. Given that the LLM-wrapper calculates the similarity between each proposal from the LVM and the input reference expression, it is generally expected that a greater number of entities would pose a greater challenge. Hence, I hope that a more in-depth study regarding the quantity of entities, particularly with respect to the inclusion of the same category, could be included in the revision.

---

> ### Author Response · Authors · 2024-12-03
> **Thank you for your feedback on the rebuttal**
>
> Thank you for your feedback and suggestion for a deeper analysis of entity counts within queries.
>
> We provide initial results on RefCOCOg val, showing that performance remains robust as the number $K$ of words in queries increases:
>
>
>  | $K$ range | # of samples | P@1    |
>  | --------- | ------------ | ------ |
>  | 1 - 5     | 1165         | 77.26  |
>  | 6 - 10    | 2531         | 78.39  |
>  | 11 - 15   | 968          | 76.75  |
>  | 16 - 20   | 193          | 79.79  |
>  | 21 - 25   | 30           | 86.67  |
>  | 26 - 35   | 9            | 100.00 |
>
> _Table R.14. P@1 of LLM-wrapper (adapting Florence-2 zero-shot with Llama3 8B) on RefCOCOg val, grouped by $K$._
>
> We will expand this analysis using HC-RefLoCo, which contains longer and more complex expressions, and refine our study to assess the impact of multiple entities or bounding boxes from the same category in the final paper.

---

### Official Review · Reviewer_f3st · 2024-11-04

**Soundness:** 2
**Presentation:** 3
**Contribution:** 2
**Rating:** 6
**Confidence:** 4

**Summary:**

Fine-tuning VLMs requires users to access the architecture, weights, and gradients of models, which can be difficult due to model privacy. This paper focuses on exploring the "black-box" adaptation of VLMs on referring expression comprehension (REC), which means only calling VLMs through forward functions. They propose LLM-wrapper, which uses the reasoning ability of LLMs to select the best matching box generated by VLMs.
In LLM-wrapper, the VLMs first generate bounding box candidates and these boxes are transferred into text prompts including the box coordinates, labels and/or prediction scores. Next, these prompts are provided to fine-tune the LLMs using LoRA. Finally, these LLMs are capable of reasoning about the image using these prompts and selecting the best matching box from the prompts. In the experimental section, they conduct experiments on three REC benchmarks. For VLMs, they choose Grounding-Dino and Florence-2, and for LLMs they choose Mixtral and Llama3. With LLM-wrapper, they show the LLMs can outperform the zero-shot VLMs by a large margin. Moreover, they find that REC performances can be further boosted by ensembling outputs from different VLMs. They also show that LLM-wrapper can be fine-tuned on one VLM and transferred to another VLM.

**Strengths:**

1. This approach does not need to access the original parameters of VLMs, but can still get benefits from them. In the case that the VLMs are closed-source, LLM-wrapper can be useful.
2. Finetuning LLMs using LoRA reduces the number of training parameters.
3. LLM-wrapper trained using one specific VLM can learn general spatial and semantic knowledge, and therefore can be used during inference with another VLM. For some expensive VLMs, with such knowledge transferring, it is not necessary to re-train another LLM-wrapper.

**Weaknesses:**

1. In table 3, for Grounding-Dino, white-box experiments are conducted using SwinB, but both zero-shot baselines and LLM-wrapper experiments are conducted using SwimT. It would be better if the authors could provide White-box results for SwinT as well.
2. In table 5, the last two rows show the results on 300 samples, but for the first four rows, the results are reported on the whole val/test sets. I would like to see the improvements when transferring from GDrec to Flo2 and from Flo2 to GDrec on the same 300 samples.
3. LLM-wrapper still uses the bounding box annotations. For REC, models with a similar number of parameters fine-tuned using the same amount of bounding boxes can even get better results.

**Questions:**

Are the results reported on different benchmarks coming from LLMs fine-tuned on corresponding training data? For example, in table 1, when the LLM-wrapper gets 78.12 on RefCOCOg val-umd, is the Llama3 fine-tuned using only the bounding boxes and images from RefCOCOg or is the Llama3 fine-tuned using samples coming from all three benchmarks?

---

> ### Author Response · Authors · 2024-11-21
> **Answer Part 1/2 to Reviewer f3st**
>
> **Answer Part 1/2**
>
> We thank the reviewer for the insightful feedback and comments.
>
> **R-f3st-W1:** *White-box experiments use SwinB, but zero-shot baselines and LLM-wrapper experiments use SwinT.*
>
> **R-f3st-A1:** We agree that using different backbones introduces variability between the two sets of experiments. However:
> - SwinT (not trained for REC) and SwinB (trained for REC) are the only two publicly available models and checkpoints (L.260-261) from Grounding-DINO's [official code](https://github.com/IDEA-Research/GroundingDINO?tab=readme-ov-file#luggage-checkpoints).
> - Our goal is not to compare directly with white-box fine-tuning, as we are working in a black-box setting (L.27, L.64, L.88, L.352). Moreover, comparing our results to the upper-bound performance of white-box fine-tuning is unfavorable to us, as we apply our method to the smaller backbone (SwinT).
> - Despite using the smaller backbone, after black-box fine-tuning with LLM-wrapper, we can achieve performance that is close to the white-box fine-tuning on the larger backbone (SwinB) for RefCOCOg (L.352-356).
>
> **R-f3st-W2:** *Results of the Transfer experiments on 300 samples (Table 5).*
>
>
> **R-f3st-A2:** As requested, we have added results for the transfer experiments (Florence-2 → GDrec, GDrec → Florence-2) evaluated on the same 300-sample subset used for GD-1.5. These results are reported in the column *P@1 Val (Subset 300)* in the _Table R.5_ below and align directly with Table 5 of the paper. The new results are marked with `#`.
>
> | Adaptation           | VLM (FT) | VLM (Infer) | P@1 Val (Full) | P@1 Val (Subset 300) | P@1 Test |
> | -------------------- | -------- | ----------- | -------------- | -------------------- | -------- |
> | None (zero-shot VLM) | None     | GDrec       | 67.61          | 66.00 #              | 68.37    |
> | LLM-wrapper          | Flo2     | GDrec       | 73.90          | 74.00 #              | 73.45    |
> | None (zero-shot VLM) | None     | Flo2        | 68.28          | 71.00 #              | 66.90    |
> | LLM-wrapper          | GDrec    | Flo2        | 73.86          | 75.33 #              | 73.03    |
> | None (zero-shot VLM) | None     | GD-1.5      | --             | 47.67                | --       |
> | LLM-wrapper          | GDrec    | GD-1.5      | --             | 76.67                | --       |
>
> _Table R.5. Results of LLM-wrapper when using different VLMs’ outputs during fine-tuning and inference, in P@1 on RefCOCOg ‘umd’ splits._
>
> We observe that results on the 300-sample subset are consistent with those on the full RefCOCOg validation set. This shows that the small 300-subset is a rather good proxy of the dataset's overall difficulty, and that the performance gains of GD-1.5 adapted with LLM-wrapper are indeed significant.
> We will include these results in the revised paper.
>
> **R-f3st-W3:** *For REC, models with a similar number of parameters fine-tuned using the same amount of bounding boxes can even get better results.*
>
>
> **R-f3st-A3:** LLM-wrapper relies on bounding box annotations, and while models fine-tuned with such annotations often perform better, this is only feasible in a white-box setting where access to the model’s architecture, weights, and training expertise is available. LLM-wrapper, by contrast, addresses the black-box scenario, where such access is not possible.
>
> Training a model from scratch using REC data and annotations is another option, but this approach requires expertise in model design and would likely fall short of achieving competitive performance. Indeed, recent SOTA REC models benefit from training on large-scale, diverse datasets such as Objects365 and Visual Genome, which include hundreds of thousands of images and complementary tasks. This rich training environment provides robustness and generalization, which cannot be achieved with REC data alone.
>
> Instead, we believe that leveraging off-the-shelf generalist VLMs is a practical approach, since these models benefit from the scale and task diversity provided by their pre-training datasets. However, without access to their internal architecture or training parameters, we explore a black-box adaptation approach in this work. LLM-wrapper effectively adapts such generalist VLMs for REC, offering a practical and resource-efficient solution for restricted-access scenarios.

---

> ### Author Response · Authors · 2024-11-21
> **Answer Part 2/2 to Reviewer f3st**
>
> **Answer Part 2/2**
>
>
> **R-f3st-Q1:** *Are the reported results for dataset X obtained with LLM fine-tuning on corresponding training set for X?*
>
> **R-f3st-AQ1:** The results reported in all experiments are obtained using LLM-wrapper fine-tuned exclusively on the corresponding training data for each benchmark, not on a combined set of samples from all three benchmarks. We will clarify this point in the revised manuscript.
>
> In addition, this question inspired us to explore LLM-wrapper's zero-shot domain transfer capabilities, across datasets.  Using the Florence-2 model adapted with LLM-wrapper on RefCOCOg-train, we evaluate its performance on RefCOCO and RefCOCO+ val/test sets. The results (_Table R.6_) show performance gains of 12.7/11.9 points on RefCOCO val/test and 7.3/6.6 points on RefCOCO+ val/test compared to zero-shot Florence-2. Although slightly below black-box fine-tuning directly on target datasets, these results highlight LLM-wrapper's ability to learn transferable knowledge. New results compared to the original paper are marked with #.
>
> | Method      | VLM        / LLM    | Training Data    | Eval Data             | P@1 (val) | P@1 (test) |
> | ----------- | ------------------- | ---------------- | --------------------- | --------- | ---------- |
> | LLM-wrapper | Florence-2 / Llama3 | RefCOCO (train)  | RefCOCO (val & test)  | 70.19     | 70.00      |
> | LLM-wrapper | Florence-2 / Llama3 | RefCOCOg (train) | RefCOCO (val & test)  | 69.00  #  | 68.88 #    |
> | 0-shot VLM  | Florence-2 / NA     | None             | RefCOCO (val & test)  | 56.32     | 57.01      |
> | LLM-wrapper | Florence-2 / Llama3 | RefCOCO+ (train) | RefCOCO+ (val & test) | 61.95     | 61.87      |
> | LLM-wrapper | Florence-2 / Llama3 | RefCOCOg (train) | RefCOCO+ (val & test) | 61.00   # | 61.07 #    |
> | 0-shot VLM  | Florence-2 / NA     | None             | RefCOCO+ (val & test) | 53.71     | 54.43      |
>
> _Table R.6. Results of LLM-wrapper in P@1 when using different datasets for training and evaluation, compared to our original setting (same dataset from training and evaluation)._
>
> This transferability reduces the need for multiple fine-tuning datasets and training processes. It also enables LLM-wrapper to be applied to datasets without training splits. For instance, as detailed in our response **R-DAad-A5** to reviewer DAad, we used an LLM-wrapper trained on RefCOCOg to evaluate our method on the proposed HC-RefLoCo benchmark, which only has validation and test splits. These findings will be added to the revised paper.

---

> ### Author Response · Authors · 2024-11-26
> **Feedbacks were incorporated in the revised paper**
>
> Dear Reviewer,
>
> As promised, we have incorporated your feedback and added clarifications and results to the revised paper, where changes are marked in blue. Below, we point to the specific sections where the changes addressing your remarks and questions have been implemented.
>
> **R-f3st-A2:** We have revised Table 5 and integrated our answer in Section 4.3 (Paragraph: "Transferring a trained LLM-wrapper to a new VLM").
>
>
> **R-f3st-AQ1:** Our answer has been integrated to the revised paper, with a clarified Section 4.2 (L.300-301), a new Table 6 and a new paragraph at the end of Section 4.3 (Paragraph: "Transferring a trained LLM-wrapper to new datasets.").
>
> As the author-reviewer discussion period continues, we would greatly value your feedback on whether our responses have sufficiently addressed your concerns.
> If you have any further comments, please feel free to share them with us.
>
> Best regards,

---

> > ### Comment · Reviewer_f3st · 2024-11-30
> > **Thanks**
> >
> > Thank you for providing the additional experimental results and the explanations. They resolve some of my concerns. I decide to raise slightly my score. This proposed method is useful for the black-box setting, even though I am concerned about the black-box setting itself. This setting is limited for the REC task since there are many open-sourced VLMs that can be fine-tuned on the REC datasets and get better performances. The most interesting setting from this paper is training LLM-wrapper on one dataset and testing on another one, or even testing on one dataset that does not have training data.

---

> ### Author Response · Authors · 2024-12-03
> **Thank you for your feedback on the rebuttal**
>
> Thank you for your response and for highlighting what you find most interesting: LLM-wrapper zero-shot transfer ability, enabled by its black-box and semantic-aware adaptation, which avoids the overfitting of white-box fine-tuning and leads to better generalization.
>
> We understand the concern about black-box fine-tuning given the availability of open-source models adapted for REC. While LLM-wrapper's true strength lies in adapting VLMs that are not pre-adapted for the task, it also complements existing white-box adaptation methods. As shown in Appendix A, LLM-wrapper achieves up to +2.4 P@1 improvements even on VLMs already fine-tuned for REC.

---

### Official Review · Reviewer_q6Km · 2024-11-06

**Soundness:** 3
**Presentation:** 3
**Contribution:** 2
**Rating:** 6
**Confidence:** 4

**Summary:**

The paper presents LLM-wrapper, a method for black-box adaptation of VLMs for the Referring Expression Comprehension (REC) task. It leverages the reasoning capabilities of LLMs to process and select the best-matching object proposals generated by VLMs like Grounding-DINO (GD) by converting VLM outputs (bounding boxes, labels, scores) into natural language prompts.

**Strengths:**

- The paper introduces a novel method, LLM-wrapper, for adapting VLMs in a black-box manner, which is an interesting way to sidestep the need for model-specific fine-tuning which involves access to model weights or architectures.
- The paper compares LLM-wrapper with baseline zero-shot and fine-tuned VLMs across 3 different REC dataset variants.
- LLM-wrapper can ensemble and transfer across different VLMs, which is an interesting future area of research.
-  LLM-wrapper requires modest resources (e.g., a single 40GB-A100 GPU)
- The paper includes thorough ablation studies on the number of training samples used for fine-tuning, comparing zero-shot LLM-wrapper with fine-tuned LLM-wrapper, showing the transferability of LLM-wrapper trained on one VLM's outputs to another VLM or ensembling VLMs, etc.
- The paper shows improvements over zero-shot, but marginally closes some gap from finetuning. Still, there is significant room for improvement which is a good direction for future work.

**Weaknesses:**

- The method requires constructing new training data to fine-tune the LLM, which partially defeats the purpose of a black-box approach that aims to minimize resource and data preparation efforts.
- The paper lacks comparisons with existing black-box adaptation methods or prompting strategies from the literature, e.g., prompting methods that were discussed in the related work section.
- There is noticeable performance variability depending on the choice of LLM or more likely the number of parameters finetuned (e.g., Llama 3 4.2% vs. Mixtral 0.24%). A deeper analysis of why these differences occur is needed, e.g., by fine-tuning fewer parameters in Llama and comparing Llama 4.2% and Llama 0.42% fine-tuned parameters.
- There is limited discussion on how well the LLM adheres to producing only the best box as the output and whether there are cases where the LLM generates unexpected or incorrect responses. More generally, the paper does not provide enough detail on failure cases where the LLM-wrapper may not perform well.
- How far are results from SOTA performance on RefCOCO REC?

Minor comment:
- The paper is too verbose in many places, e.g., section 3.4 seems unnecessary as the benefits have been explained well in the previous sections. Instead, it would be nice to provide more analysis with SoTA baselines, more diverse qualitative examples, limitations, etc.

**Questions:**

- Can you explain in more detail how GroundingDino (GD) is used as zero-shot method for REC task, given that GD outputs bounding boxes that cover various objects in an image but cannot directly perform the reasoning needed to decide which box matches a complex query?

- Is the number of fine-tuned parameters an important aspect in achieving better performance? How does Llama with 0.42% parameters tuned performs in comparison with the Llama version in the paper (with 4.2% parameters fine-tuned)?

- Does the LLM always adhere to the generated bbox index?

---

> ### Author Response · Authors · 2024-11-21
> **Answer Part 1/3 to Reviewer q6Km**
>
> **Answer Part 1/3**
>
> We thank the reviewer for the insightful feedback and comments.
>
> **R-q6Km-W1:**.*The method requires constructing new training data to fine-tune the LLM, which partially defeats the purpose of a black-box approach that aims to minimize resource and data preparation efforts.*
>
> **R-q6Km-A1:** Indeed, LLM-wrapper requires REC data to fine-tune the LLM.  However, we respectfully disagree that this contradicts the black-box paradigm. The black-box approach focuses on adapting models without access to their internal architecture or parameters, rather than on minimizing resource and data preparation efforts.
>
> Nonetheless, LLM-wrapper remains resource-efficient and practical for several reasons:
> * *Straightforward data preparation:* As described in Section 3.2, constructing training data simply involves converting VLM outputted boxes into textual coordinates, making the process lightweight.
> * *Efficient training:* The training process is highly efficient, taking less than 2 hours on a single A100 GPU.
> * *Adaptability to new VLMs:* As shown in Section 4.3 (L.408-419), a trained LLM-wrapper can be applied to different VLMs without additional fine-tuning.
>
> Additionally, LLM-wrapper generalizes well to new datasets in a zero-shot manner, eliminating the need for creating specific training data for each new target dataset. For instance, in a new experiment, we evaluate LLM-wrapper on the HC-RefLoCo dataset, which includes longer and more complex referring expressions than RefCOCO/+/g. Since HC-RefLoCo lacks a training set, we *transfer* an LLM-wrapper trained on RefCOCOg to this dataset, following a protocol similar to Section 4.3.
> Our results demonstrate a significant +18.8 (resp. +18.4) P@1 improvement over zero-shot Florence-2 on the val (resp. test) split, showcasing LLM-wrapper's strong transferability and capacity for complex reasoning:
>
> | Method      | VLM       / LLM     | Training Data  | P@1 (HC-RefLoCo val) | P@1 (HC-RefLoCo test) |
> | ----------- | ------------------- | -------------- | -------------------- | --------------------- |
> | 0-shot VLM  | Florence-2          | None           | 48.64                | 48.87                 |
> | LLM-wrapper | Florence-2 / Llama3 | RefCOCOg train | 67.40                | 67.26                 |
>
> _Table R.1. LLM-wrapper's dataset transfer performance, evaluated on the HC-RefLoCo benchmark (with prior fine-tuning on RefCOCOg), vs zero-shot Florence 2._
>
>
> We also have new results, included in our response **R-f3st-AQ1** to Reviewer f3st, demonstrating further successful transfers (RefCOCOg → RefCOCO and RefCOCOg → RefCOCO+), which align with these findings. Overall, these results show a strong LLM-wrapper’s transferability, significantly reducing the need for new training data for each target application.
>
> We will clarify these points and include these results in the revised paper.
>
> **R-q6Km-W2:** *Comparisons with existing black-box adaptation methods or prompting strategies from the literature.*
>
>
> **R-q6Km-A2:** The related work section of our paper discusses existing black-box adaptation methods. However, these methods are unsuitable for our scenario for the following reasons:
>
> * *Non-applicability of prompting strategies:* Methods, such as those by Ouali et al. (2023), Yu et al. (2023), and Liu et al. (2024b), focus on *CLIP-based* models that rely on *prompt templates* or require access to *intermediate representations of the model*. Our work does not use prompt templates as CLIP does, since our target VLMs (e.g., Grounding-DINO and Florence-2) perform open-vocabulary detection and do not require such inputs. Moreover, the need to access intermediate representations of the model does not qualify as a fully black-box setting.
>
> * *We focus on task adaptation, not domain shifts:* Oh et al. (2023) modify image inputs for visual domain adaptation. However, this approach addresses domain gaps, not task adaptation like adapting a generalist VLM for REC.
>
> Our method is specifically designed to adapt VLMs to the REC task, in a true black-box manner, without requiring intermediate representations, gradients, or multiple API calls. To the best of our knowledge, it is the first proposal aimed at this goal, and we could not identify any baseline method in the literature for direct comparison.

---

> ### Author Response · Authors · 2024-11-21
> **Answer Part 2/3 to Reviewer q6Km**
>
> **Answer Part 2/3**
>
> **R-q6Km-W3:** *Performance variability depending on the choice of LLM and number of finetuned parameters.*
>
> **R-q6Km-A3:** Our ablation study on LoRA's rank $r$, presented in Figure 4(b), evaluates P@1 across various $r$ values. Since the number of trainable parameters scales linearly with LoRA's rank, this analysis indirectly illustrates how the number of trainable parameters affects performance.
> To clarify this aspect, and to answer more precisely the reviewer’s comment, we conduct a more detailed analysis of the impact of the number of fine-tuning parameters. LLM-wrapper's results using Llama3 8B and other LLMs (applied to Grounding-DINO (GD) and evaluated on RefCOCOg val-umd), are given in the _Table R.2_ below, showing the relationship between the number of trainable parameters and performance. We also report the original performance of the respective LLMs on the standard MMLU benchmark and HellaSwag, a benchmark for LLM commonsense reasoning evaluation.
>
>
> | LLM              | Rank ($r$) | Trainable Params | Total Params | Trainable Params % | P@1   | LLM MMLU | LLM HellaSwag |
> | ---------------- | ---------- | ---------------- | ------------ | ------------------ | ----- | -------- | ------------- |
> | **Llama3 8B**    | 64         | 176M             | 8.2B         | 2.15%              | 77.72 | 66.6     | 82            |
> | **Llama3 8B**    | 128        | 352M             | 8.4B         | 4.20%              | 78.12 | 66.6     | 82            |
> | **Llama3 8B**    | 192        | 528M             | 8.6B         | 6.18%              | 77.70 | 66.6     | 82            |
> | **Mixtral 8×7B** | 128        | 113M             | 46.8B        | 0.24%              | 77.57 | 70.6     | 84.4          |
> | **Gemma-2 9B**   | 128        | 465M             | 9.7B         | 4.79%              | 74.12 | 71.3     | 81.9          |
> | **Gemma-2 2B**   | 128        | 199M             | 2.8B         | 7.08%              | 70.51 | 52.2     | 72.9          |
>
> _Table R.2. Analysis of the impact of the number of trainable parameters on LLM-wrapper's performances._
>
>  As part of our analysis, we computed the Pearson correlation between the P@1 scores and other variables from _Table R.2_:
>
> | Variable                         | Pearson Correlation with P@1 |
> | -------------------------------- | ---------------------------- |
> | LLM Total Parameters (count)     | 0.38                         |
> | LLM Trainable Parameters (count) | 0.08                         |
> | LLM Trainable parameters (%)     | 0.37                         |
> | LLM MMLU score                   | 0.73                         |
> | LLM HellaSwag score              | 0.88                         |
>
> _Table R.3. Pearson Correlation with P@1 for various variables._
>
> _Table R.3_ shows that the absolute number of fine-tuned parameters has a loose correlation with performance (pearson=0.08). This supports findings from _Table R.2_ showing the limited impact of the number of parameters, such as:
> - Variations in LoRA's rank $r$ do not have a large effect on the P@1 score of Llama3 8B (cf _Table R.2_, rows 1 to 3).
> - Llama3 8B (rank=64, 176M trainable parameters) and Mixtral 8×7B (rank=128, 113M trainable parameters) yield very similar P@1 scores (77.72 vs. 77.57), despite Llama3 8B fine-tuning a much higher percentage (2.15% vs. 0.24%) of parameters. The similarity of results could then be due to architectural differences, with Mixtral being a Mixture of Experts model.
> - Gemma-2 9B (rank=128, 465M trainable params) underperforms compared to Llama3 8B (rank=64, 176M trainable params), even though its setting includes fine-tuning more than double the number of parameters.
>
> These findings suggest that the architecture of the LLM and training specifics may have more influence on performances.
>
> Furthermore, our analysis finds a strong correlation between LLM-wrapper's P@1 scores and the LLM's performances on reasoning tasks, with a Pearson correlation of 0.88 with HellaSwag. This indicates that LLM-wrapper's performance is mostly dependent on the original performance of the LLM on reasoning tasks.
>
> In conclusion, the number of fine-tuned parameters has a small impact, but the LLM's architecture and other training factors play a more significant role regarding performance. We plan to extend this analysis with further studies on the impact of LoRA's rank values and of the number of fine-tuned parameters for other LLMs. We will include these findings in the revised paper.

---

> ### Author Response · Authors · 2024-11-21
> **Answer Part 3/3 to Reviewer q6Km**
>
> **Answer Part 3/3**
>
> **R-q6Km-W4:** *Failure case analysis. Does the LLM adhere to producing only the best box as the output?*
>
>
> **R-q6Km-A4:**
> Indeed, a failure case arises when the LLM does not adhere to the rule of producing only the index of a candidate bounding box as output. In such cases, the LLM may output non-integer values or indices that are out of range of the candidate boxes' list. However, these issues are very rare (<0.5%). In _Table R.4_ below, we report the percentage of invalid outputs, when Florence-2 is adapted with LLM-wrapper, with either Mixtral or Llama3. In these rare failure scenarios, we use a simple fallback strategy: the best-ranked box from the zero-shot VLM is used as prediction.
>
> |                        | RefCOCOg Val | RefCOCOg Test | RefCOCO Val | RefCOCO Test | RefCOCO+ Val | RefCOCO+ Test |
> | ---------------------- | ------------ | ------------- | ----------- | ------------ | ------------ | ------------- |
> | LLM-wrapper w/ Mixtral | 0.02%        | 0.05%         | 0.03%       | 0.10%        | 0.13%        | 0.09%         |
> | LLM-wrapper w/ Llama3  | 0.02%        | 0%            | 0.41%       | 0.35%        | 0.16%        | 0.18%         |
> _Table R.4. Percentages of invalid LLM outputs._
>
> The main failure case of LLM-wrapper, described in L.497-500, occurs when bounding box coordinates and labels alone are insufficient to localize certain referring expressions that require additional visual cues. For example, in Figure 2f, correctly answering the query "White plane in the front" requires knowing the orientation of the planes.
> We will include more such examples, and the invalid box index failure case, in the revised paper.
>
>
> **R-q6Km-W5:** *How far are results from SOTA performance on RefCOCO REC?*
>
>
> **R-q6Km-A5:** At the time of writing the rebuttal, the state-of-the-art (SOTA) performance on RefCOCO/+/g REC remains Florence-2. To be more precise, there are two distinct SOTAs:
> * White-box adaptation setting: The SOTA performance occurs when Florence-2 is trained on RefCOCO/+/g data and the REC task, along with other tasks and datasets. For example, the SOTA on RefCOCOg (val-umd) is 90.3 (reported in Table 3, RefCOCOg val-umd). Our work does not aim to replace or outperform this setting, as it focuses on the black-box paradigm (L.27, 64, 88, 352). However, when LLM-wrapper is used to adapt this version of Florence-2, it slightly improves performance to 90.4 (see our answer **R-DAad-A3** to reviewer DAad for details).
> * Zero-shot setting: When Florence-2 is not trained on the REC task with the RefCOCO/+/g datasets, the SOTA is 68.3 (Table 3) on RefCOCOg (val-umd). We adapt this model with LLM-wrapper, with black-box fine-tuning, and achieve a performance of 77.9 (Table 3).
>
>
>
> **R-q6Km-W6:** *Minor comment: The paper is too verbose in many places, e.g., section 3.4 seems unnecessary as the benefits have been explained well in the previous sections. Instead, it would be nice to provide more analysis with SoTA baselines, more diverse qualitative examples, limitations, etc.*
>
>
> **R-q6Km-A6:** Thank you for this suggestion. We will carefully review the paper again, and Section 3.4 in particular, to ensure conciseness. This adjustment will allow us to allocate space for additional analysis, including:
> * New results on zero-shot dataset transfer, VLM transfer, HC-RefLoCo dataset, Kosmos-2 VLM,
> * New analyses on the influence of the number of entities in the query, of the LLM reasoning abilities and of the number of trainable parameters, on the P@1 performance,
> * More diverse qualitative examples, including success and failure cases,
> * Clarifications adressing reviewers' feedbacks.
>
>
> **R-q6Km-Q1:** *How is GroundingDino (GD) used as zero-shot method for REC task?*
>
>
> **R-q6Km-AQ1:** In our zero-shot experiments, we use GroundingDino (GD) by selecting the bounding box that has the highest score with respect to any token in the query, from the 900 proposals generated by GD (L.255-256). However, this approach sometimes selects boxes based on parts of the query unrelated to the main object of interest, such as other nouns in the sentence. To mitigate this, we introduce a straightforward alternative, GDrec, which first identifies the query's subject and then selects the bounding box that scores best against it (L.254-257).
> We will rephrase and clarify in the revised paper.
>
> **R-q6Km-Q2:** *Is the number of fine-tuned parameters an important aspect in achieving better performance?*
>
> Please see our answer above (**R-q6Km-A3**)
>
> **R-q6Km-Q3:** *Does the LLM always adhere to the generated bbox index?*
>
> Please see our answer above (**R-q6Km-A4**)

---

> ### Author Response · Authors · 2024-11-26
> **Feedbacks were incorporated in the revised paper**
>
> Dear Reviewer,
>
> As promised, we have incorporated your feedback and added clarifications and results to the revised paper, where changes are marked in blue. Below, we point to the specific sections where the changes addressing your remarks and questions have been implemented.
>
> **R-q6Km-A1:** Our answer has been integrated to the revised paper, with the new Table 6 and new paragraph at the end of Section 4.3 (Paragraph: "Transferring a trained LLM-wrapper to new datasets.").
>
> **R-q6Km-A3:** Our answer has been integrated to the revised paper with a clearer Section 4.4 and the new Appendix B.1.
>
> **R-q6Km-A4:** Our answer has been integrated to the revised paper with the new Appendix C, referred to in Section 4.1.3.
>
> **R-q6Km-A5:** Our answer has been integrated to the revised paper with the new Appendix A, referred to in Section 4.2.
>
> **R-q6Km-A6:** We have removed Section 3.4 and integrated its key content into the introduction. A discussion on the limitations has been added to the conclusion. Additional results have been included in Section 4.3, highlighting zero-shot dataset transfer, and in Appendix A, demonstrating black-box adaptation for already fine-tuned VLMs. Additional ablations on variables impacting performances are performed in Appendix B. An analysis on failure cases was added in Appendix C and additional qualitative examples are displayed in Appendix D.
>
> **R-q6Km-AQ1:** Our answer has been integrated to the revised paper with a clearer Section 4.1.2
>
> As the author-reviewer discussion period continues, we would greatly value your feedback on whether our responses have sufficiently addressed your concerns.
> If you have any further comments, please feel free to share them with us.
>
> Best regards,

---

> > ### Comment · Reviewer_q6Km · 2024-11-29
> > **Thank you for the responses**
> >
> > Thank you for the detailed and well-structured responses to my comments. I appreciate the effort in addressing my concerns and providing additional experiments. The detailed analysis of the impact of trainable parameters along with additional ablation results is a great addition to the paper. For these reasons, I have slightly raised my score.
> >
> > However, it would be another great addition to show that the method generalizes well to new datasets apart from the RefCOCO family as I believe other reviews have pointed out too. And given the rise of pixel-grounding LLMs that utilize RefCOCO as an evaluation benchmark, e.g., GLaMM, PixelLM, LISA, etc., the technical novelty of using a LoRa adapter module and converting VLM outputted boxes into textual coordinates seems limited.
> >
> > Finally, the reasoning behind why existing black-box adaptation methods are not applicable as baselines is a bit unconvincing to me. Even if these require prompt templates or are semi-black-box methods, it is unclear why they cannot be reported as baselines in the paper, with an explicit discussion of their drawbacks vs. performance comparisons. As an alternative, recent pixel-grounding VLM baselines would show how the method compares to state-of-the-art VLMs in terms of raw REC performance before adaptation.

---

> ### Author Response · Authors · 2024-12-03
> **Thank you for your feedback on the rebuttal**
>
> Thank you for your feedback and insights.
>
>
> `[Baselines and Pixel-grounding large multimodal models (LMMs)]`
>
> We appreciate your suggestion regarding pixel-grounding models, and we will cite these works in the revised paper. LLM-wrapper is a model-agnostic adaptation framework that emphasizes object-level reasoning, complementing the pixel-level reasoning of end-to-end systems like GLaMM or PixelLM. We will include additional baselines for upper-bound comparison, including a semi-black-box adaptation method and state-of-the-art pixel-grounding LMMs.
>
> `[More datasets]`
>
> We expanded our evaluation beyond the RefCOCO/+/g family and HC-RefLoCo by testing on the Talk2Car REC benchmark [1], which involves commands for driving scenes. This allowed us to assess generalization across a domain that differs from human-centric images and standard textual descriptions. Results are summarized below in _Table R.13_.
>
> | Adaptation                    | VLM        | Fine-tuning REC Data                    | P@1 (val)          | P@1 (test)         |
> | ----------------------------- | ---------- | --------------------------------------- | ------------------ | ------------------ |
> | None (0-shot VLM)             | Florence-2 | None                                    | 35.17              | 37.80              |
> | Classic white-box fine-tuning     | Florence-2 | Multiple datasets including RefCOCO/+/g | 37.06  **(↑1.89)** | 40.25 **(↑2.45)**  |
> | LLM-wrapper                   | Florence-2 | RefCOCOg                                | 44.63 **(↑9.46)**  | 46.47 **(↑8.67)**  |
> | LLM-wrapper                   | Florence-2 | Talk2Car                                | 60.71 **(↑25.54)** | 65.59 **(↑27.79)** |
>
> _Table R.13. Evaluation on Talk2Car (REC) with LLM-wrapper vs. zero-shot and white-box fine-tuned Florence-2 (LLM-wrapper uses Llama3 8B)._
>
> Key findings:
> - *Zero-shot transfer:* LLM-wrapper adapted on RefCOCOg improves P@1 by +9.46 (val) and +8.67 (test) compared to zero-shot Florence-2, showing generalization capabilities despite the domain gap. In contrast, white-box fine-tuning of Florence-2 on RefCOCO/+/g only slightly improves performances in the new domain with +1.89 (val) and +2.45 (test), indicating a lack of generalization.
> - *Targeted adaptation:* LLM-wrapper achieves better results when adapted on Talk2Car (train), boosting P@1 by +25.54 (val) and +27.79 (test).
>
> These results illustrate LLM-wrapper's strength in cross-domain generalization and its effectiveness as a black-box adaptation method. We will include these findings in the final paper.
>
> [1] T. Deruyttere et al., *Talk2Car: Taking Control of Your Self-Driving Car,* EMNLP/IJCNLP 2019.

---

### Author Response · Authors · 2024-11-20
**General answer to all reviewers**

We thank all reviewers for their thoughtful feedback, valuable insights and for underlining several strengths and contributions of our work.

Reviewers appreciated that LLM-wrapper is **"novel"** (q6km) and "**impressive**" (o1yY). They highlighted the value of our **black-box** setting, which avoids the need for model-specific fine-tuning expertise and direct access to VLM parameters (q6Km, f3st, DAad). The **flexibility** of LLM-wrapper was also noted, particularly its ability to *transfer* spatial and semantic knowledge from one VLM to others without retraining (q6Km, f3st, DAad), as well as its capability to *ensemble* outputs from multiple VLMs (q6Km). We are pleased that reviewers found the **experiments and ablation** studies thorough (q6Km) and acknowledged the efficiency of our method, which requires only modest resources (q6Km, f3st, DAad). The paper’s **clarity** was also appreciated, making the method and its motivations easy to understand (DAad, o1yY).

We address reviewers’ concerns and questions with direct answers to each reviewer, and we will incorporate the suggested improvements in the final version of the paper.

---

### Author Response · Authors · 2024-11-26
**Message to all reviewers**

Dear reviewers,

Thank you again for the time and effort you have invested in reviewing our work.
We addressed your concerns and questions and incorporated suggested improvements in the updated version of the paper, where changes are marked in blue.
As the author-reviewer discussion period is ongoing, we would greatly value your feedback on whether our responses have sufficiently addressed your concerns.
If you have any further comments, please feel free to share them with us.

Best regards,

---

### Meta-Review · Area_Chair_MpMQ · 2024-12-20

**Metareview:**

This paper proposed a simple yet effective approach for tackling the referring expression task. Concretely, the paper proposed to attach a LLM to the output of the VLM model. The VLM model takes the input image and the text, and output the bounding box and the bounding box description. Based on the input text and the bounding box description, the LLM generates the output. The LLM can be further LoRA finetuned to improve its performance.

Strength:
1. The idea is simple yet effective.
2. This idea provides a way to further improve the close source model's output.
3. The proposed approach achieves good performance.

All the reviewers vote support for this paper, I would recommend accept.

**Additional Comments On Reviewer Discussion:**

The rebuttal mainly focuses on the inference cost, additional training data, and the impact of VLM/LLM on the final performance. The author added an additional limitation section to acknowledge the inference cost. The author also provided more ablation study to justify the effectiveness of the approach.

---

### Decision · Program_Chairs · 2025-01-22

Accept (Poster)